# The unifying catalytic mechanism of the RING-between-RING E3 ubiquitin ligase family

Xiangyi S. Wang[1,2,3], Thomas R. Cotton [1,2,3], Sarah J. Trevelyan[1,2], Lachlan W. Richardson [1,2], Wei Ting Lee[1], John Silke [1,2] & Bernhard C. Lechtenberg [1,2] ✉

The RING-between-RING (RBR) E3 ubiquitin ligase family in humans comprises 14 members and is defined by a two-step catalytic mechanism in which ubiquitin is first transferred from an E2 ubiquitin-conjugating enzyme to the RBR active site and then to the substrate. To define the core features of this catalytic mechanism, we here structurally and biochemically characterise the two RBRs HOIL-1 and RNF216. Crystal structures of both enzymes in their RBR/E2-Ub/Ub transthiolation complexes capturing the first catalytic step, together with complementary functional experiments, reveal the defining features of the RBR catalytic mechanism. RBRs catalyse ubiquitination via a conserved transthiolation complex structure that enables efficient E2-to-RBR ubiquitin transfer. Our data also highlight a conserved RBR allosteric activation mechanism by distinct ubiquitin linkages that suggests RBRs employ a feed-forward mechanism. We finally identify that the HOIL-1 RING2 domain contains an unusual Zn2/Cys6 binuclear cluster that is required for catalytic activity and substrate ubiquitination.

E3 ubiquitin ligases are the effector enzymes of the ubiquitination machinery, and can be broadly divided into three main families, based on their catalytic mechanism[1,2]. RING (Really Interesting New Gene) E3s mediate direct ubiquitin (Ub) transfer from an E2-Ub thioester conjugate to the substrate protein. RING domains bind the E2 and induce a closed E2-Ub conformation that is essential for the attack of the E2-Ub thioester bond by the substrate nucleophile (generally a lysine side-chain) in an aminolysis reaction[3]. HECT (Homologous to E6-AP C-Terminus) E3 ligases are structurally distinct from RING E3s and catalyse substrate ubiquitination in a two-step reaction[4]. Ub is first transferred from the E2 active site cysteine to a catalytic cysteine in the HECT C-lobe in a transthiolation reaction. In a subsequent step, Ub is transferred from the HECT E3 to the substrate. RING-between-RING (RBR) E3 ligases contain three zinc-binding domains termed RING1, in-between RING (IBR) and RING2 (collectively called the RBR module)

and are described as RING/HECT hybrid E3 ligases[5]. The RBR RING1 domain features a RING fold and binds the E2 analogous to the RING E3s[6]. However, RBRs do not induce the essential closed E2-Ub conformation, but instead stabilise an open E2-Ub conjugate conformation and align the active site cysteine in their RING2 domain with the E2-Ub thioester for Ub transfer from the E2 to the RBR E3 in a transthiolation reaction[6,7]. In a subsequent step, Ub is transferred from the RBR RING2 active site to the substrate in an aminolysis reaction, analogous to the mechanism of the structurally distinct HECT E3 ligases.

Given the critical role of ubiquitination in cellular homeostasis and signalling, it is unsurprising that RBR E3 ligase activity is often subject to strict regulation. Regulation via multiple mechanisms ensures substrate ubiquitination occurs in appropriate spatio-temporal contexts[8,9]. Since the RBR RING/HECT-hybrid mechanism was first proposed by the Klevit lab a decade ago[5], the field has made

[1]Ubiquitin Signalling Division, The Walter and Eliza Hall Institute of Medical Research, Parkville, Victoria 3052, Australia. [2]Department of Medical Biology, The University of Melbourne, Parkville, Victoria 3010, Australia. [3]These authors contributed equally: Xiangyi S. Wang, Thomas R. Cotton. ✉e-mail: lechtenberg.b@wehi.edu.au

vast progress in understanding the catalytic mechanism of select members of the 14 human RBR E3 ligases, namely for Parkin, HHARI (ARIH1) and HOIP. Several RBR E3 ligases undergo multistep activation processes including displacement of auto-inhibitory domains and large-scale conformational changes that bring the active sites of the RING1 bound E2-Ub, and RING2 domain into reactive proximity. Parkin is activated by phosphorylation of the Parkin ubiquitin-like (UBL) domain by the kinase PINK1[10–12]. HHARI is kept autoinhibited by its C-terminal Ariadne domain and activated by interaction with NEDDy-lated cullins or phosphorylation in its Ariadne domain[13–16]. HOIP is autoinhibited by a UBA (ubiquitin associated) domain N-terminal to the RBR module and activated by interactions with its cofactors HOIL-1 or Sharpin in the linear ubiquitin chain assembly complex (LUBAC)[17–20].

Further to autoinhibition, allosteric activation by Ub or ubiquitin-like proteins (UBLs) is emerging as a common regulatory feature of RBR E3 ligases. This is best understood for Parkin, where binding of phospho-Ub (Ub phosphorylated on S65 by PINK1) to an interface between the RING1 and IBR domains is critical for activation[11,21,22]. Similarly, HHARI is activated by the UBL NEDD8 when NEDD8 is conjugated to a cullin in a HHARI/NEDD8-CUL1-RBX1 complex[14,16]. We previously showed that HOIP is activated by M1-linked, linear, di-Ub and that the allosteric binding site in HOIP is required for HOIP-mediated NF-κB activation in cells[6]. Similarly, the K63-linkage-specific RBR RNF216 is allosterically activated by K63 di-Ub, but not other di-Ub linkages[23]. While binding of di-Ub appears to stabilise or promote a conformation in which the E2-Ub conjugate can be accommodated in the RBR active site, the precise mechanism by which Ub binding at an allosteric site enhances catalysis has not been explored in detail.

The focus on just a few of the RBR family members leaves a gap in our understanding as to which of these structural and functional features are conserved within the RBR family and which are adaptations of individual enzymes to specific functions.

To identify the unifying features of the RBR catalytic mechanism, we here biochemically, biophysically, and structurally investigate the previously poorly characterised RBRs HOIL-1 and RNF216. We focus on the first catalytic step, i.e., Ub transfer from the E2 to the E3 (trans-thiolation), and present crystal structures of both RBRs captured in E2-Ub conjugate loaded transthiolation complexes with an additional allosteric Ub. Our work highlights that RBRs form structurally conserved transthiolation complexes. We postulate that RBRs are feed-forward enzymes that are specifically activated by distinct Ub linkages that are the direct product of the RBR reaction or associated with their signalling pathway. In addition to identifying these common RBR structural and mechanistic features, our work reveals unique features of HOIL-1. Most surprisingly, the HOIL-1 RING2 domain comprises a Zn2/Cys6 binuclear cluster that replaces the C-terminal half of the RING2, is required for HOIL-1 activity and substrate ubiquitination and may point to the recently reported ability of HOIL-1 to non-conventionally ubiquitinate hydroxyl groups in proteins (Ser and Thr residues) and polysaccharides[24–27].

## Results

### HOIL-1 and RNF216 are allosterically activated by distinct di-Ub species

To determine if HOIL-1, like other RBRs, is activated by specific Ub chain types, we performed an E2-Ub thioester discharge assay with the Cys-reactive E2 UbcH7 and a HOIL-1 construct comprising the RBR module and an N-terminal helix (helix-RBR, Fig. 1a). HOIL-1 shows poor E2-Ub discharge activity in the absence of an allosteric activator or with 5 μM of most di-Ub species, but its activity is strongly enhanced in the presence of M1- or K63-linked di-Ub (Fig. 1b). Di-Ub titration experiments show that M1 di-Ub ($EC_{50} = 8\,\mu M$) is more than 2-times more potent as a HOIL-1 activator than K63 di-Ub ($EC_{50} = 18\,\mu M$; Fig. 1c).

HOIL-1 contains an Npl4 zinc finger (NZF) domain N-terminal to the RBR module (Fig. 1a) that has previously been shown to specifically

bind M1-linked di-Ub[28], suggesting that it may aid in recruitment of allosteric M1-linked di-Ub. However, HOIL-1 is activated by M1-linked di-Ub independently of the NZF domain, as indicated by comparable $EC_{50}$ values for M1-linked di-Ub activation of HOIL-1 helix-RBR and NZF-RBR constructs (Fig. 1d). These results are consistent with recently published work by Kelsall et al.[24].

We further performed the E2-Ub discharge experiments with RNF216. As previously observed with the oxyester-linked UbcH5B(C85S)-Ub conjugate[23], UbcH7-Ub discharge by RNF216 helix-RBR-helix (Fig. 1a) is specifically activated by K63 linked di-Ub but no other di-Ub species (Fig. 1e), suggesting that RBR allosteric activation is independent of the E2 used.

Together with previous data on allosteric activation of the RBR ligases Parkin (pUb)[11], HOIP (M1-linked di-Ub)[6], and HHARI (NEDDy-lated cullin)[14,16], our data support the idea that allosteric activation by Ub or UBLs is a common feature of the RBR family.

### Allosteric activation enhances E2-Ub binding and E2-to-RBR transthiolation

Given our observations that allosteric Ub enhances RBR E3-mediated E2-Ub discharge, we hypothesised that allosteric Ub binding to the RBR increases binding affinity for the E2-Ub conjugate. To test this hypothesis, we assessed binding of a stable, non-reactive UbcH7(C86K)-Ub conjugate to catalytically inactive (active site Cys mutated to Ala) HOIL-1 helix-RBR (C460A) and RNF216 RBR-helix (C688A, pSer719) in the absence or presence of activating di-Ub using isothermal titration calorimetry (ITC). In the UbcH7(C86K)-Ub conjugate, the UbcH7 active site Cys is replaced by Lys that forms an isopeptide bond with the Ub C-terminus, mimicking a Ub-loaded UbcH7. We use RNF216 phosphorylated on Ser719 (pSer719) for ITC to better determine the relatively weak binding[23].

In the absence of allosteric Ub, we observe no stable binding of the UbcH7(C86K)-Ub conjugate to the HOIL-1 helix-RBR module (Fig. 2a, c). However, the UbcH7(C86K)-Ub conjugate binds tightly when HOIL-1 is preincubated with the allosteric activator M1 di-Ub ($K_D = 0.2–0.5\,\mu M$, n = 2), whereas preincubation with K48 di-Ub, which does not activate HOIL-1 in the E2-Ub discharge assay (Fig. 1b), enhances E2-Ub binding to a lesser extent ($K_D = 5.3\,\mu M$, Fig. 2a, c).

Like HOIL-1, UbcH7(C86K)-Ub conjugate binding to RNF216 RBR-helix is not detected in the absence of allosteric activator, but pre-incubation of RNF216 with the activating K63 di-Ub facilitates binding of E2-Ub with an affinity of 1.3 μM, whereas addition of K48 di-Ub had a less pronounced effect on E2-Ub binding ($K_D = 4.2\,\mu M$, Fig. 2b, c).

The ITC data, together with previous structural studies showing straightening of the RING1-IBR helix upon allosteric Ub/UBL binding[6,14,21], suggest that binding of allosteric di-Ub induces conformational changes in the HOIL-1 and RNF216 RBR modules that promote high-affinity E2-Ub binding and formation of the ternary RBR/E2-Ub/di-Ub transthiolation complex.

Having established that specific di-Ub species activate the first step of the RBR catalytic cycle (transthiolation), we next investigated whether allosteric di-Ub also activates the second (aminolysis) step of the RBR reaction by monitoring Ub discharge from the RBR E3 and substrate formation. In contrast to HOIL-1 and RNF216, the M1-linkage-specific RBR HOIP efficiently forms an RBR-Ub thioester intermediate in the absence of a native free Ub N-terminus (GPG-Ub). Subsequent addition of native Ub with a free N-terminus initiates E3-Ub discharge and M1 di-Ub formation, enabling us to independently follow both RBR reaction steps.

HOIP chain formation (Supplementary Fig. 1a) and transthiolation (Supplementary Fig. 1b) reactions are activated by M1-linked di-Ub with an $EC_{50}$ of 19.9 μM, but not by M1-linked di-Ub harbouring the I44A mutation in the canonical binding site[6]. Addition of fluorescently labelled Ub (TAMRA-Ub) with a native N-terminus to GPG-Ub-loaded HOIP RBR induces the RBR aminolysis reaction, E3-Ub discharge, and

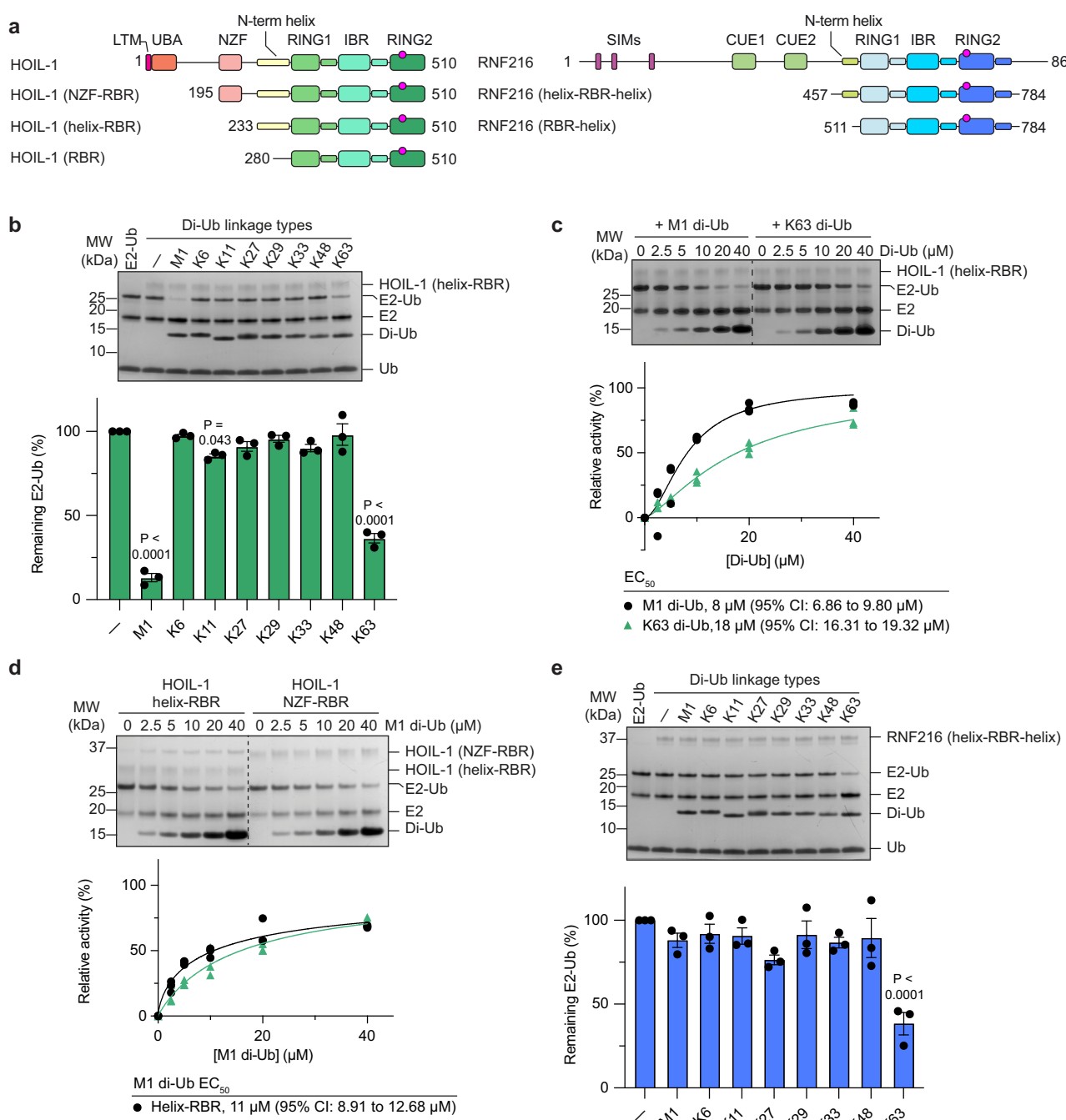

**Fig. 1 | HOIL-1 and RNF216 are allosterically activated by specific di-Ub species.** **a** Schematic showing HOIL-1 and RNF216 protein domains and constructs used in this study. The active site is indicated by a red circle. LTM, LUBAC-tethering motif; UBA, Ub-associated domain; NZF, Npl4 zinc finger domain; SIM, SUMO-interacting motif; CUE, Coupling of Ub conjugation to ER degradation domain. **b** HOIL-1 helix-RBR catalysed UbcH7-Ub discharge assay in the presence of different di-Ub species (5 µM). The top panel shows a representative Coomassie-stained SDS-PAGE gel. The bottom panel shows quantification of three independent experiments. Individual data points and mean +/− SEM shown. *P* values <0.05 compared to no activator from one-way ANOVA with Tukey's post-test are indicated. MW, molecular weight marker. **c** HOIL-1 catalysed UbcH7-Ub discharge assay in the presence of increasing concentrations of M1-linked (left) or K63-linked (right) di-Ub. The bottom panel

shows quantification of three independent experiments for $EC_{50}$ determination. Nonlinear regression curves were fitted using the [agonist] vs. normalised responses−variable slope model. CI, confidence interval **d** HOIL-1 catalysed UbcH7-Ub discharge assay with helix-RBR and extended NZF-RBR constructs. The bottom panel shows quantification of three independent experiments for $EC_{50}$ determination. Nonlinear regression curves were fitted using the [agonist] vs. normalised responses−variable slope model. **e** Same as in panel (**b**) but with RNF216 as the E3 ligase. The top panel shows a representative Coomassie-stained SDS-PAGE gel. The bottom panel shows quantification of three independent experiments. Individual data points and mean +/− SEM shown. *P* values <0.05 compared to no activator from one-way ANOVA with Tukey's post-test are indicated. Source data are provided as a Source data file.

formation of TAMRA-di-Ub independent of the addition of allosteric M1-linked di-Ub (Supplementary Fig. 1c, d). These results show that M1-linked di-Ub only accelerates the first (transthiolation) but not the second (aminolysis) step of the HOIP RBR catalytic reaction.

## RBRs form a conserved RBR/E2-Ub/Ub transthiolation complex

We further evaluated formation of the RBR/E2-Ub/di-Ub ternary complex for both HOIL-1 and RNF216 using size-exclusion chromatography coupled to multi-angle light scattering (SEC-MALS). We observe

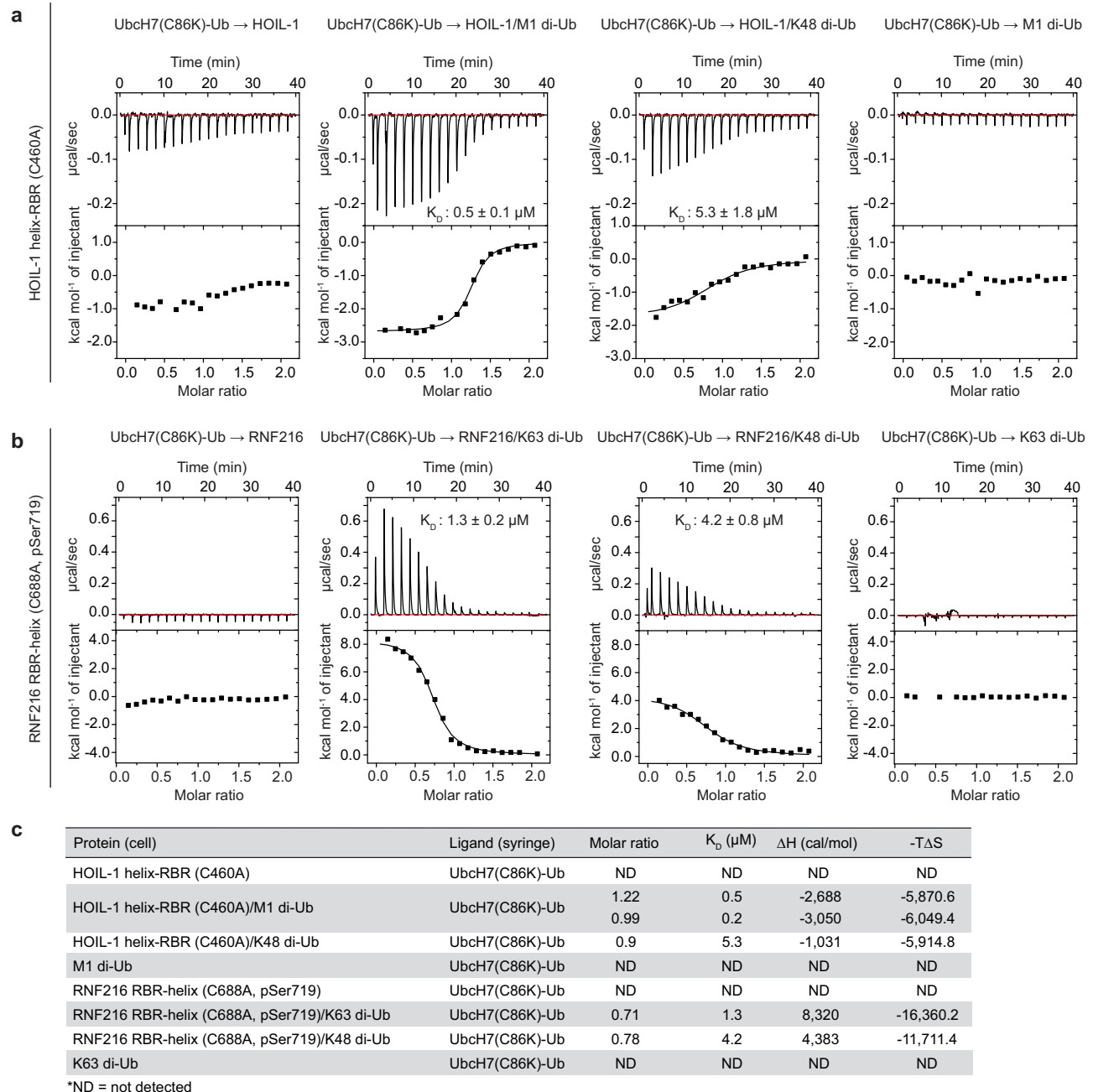

**Fig. 2 | Allosteric di-Ub enhances UbcH7-Ub binding affinity for HOIL-1 and RNF216. a** Representative isothermal titration calorimetry (ITC) of isopeptide linked UbcH7(C86K)-Ub conjugate titrated into HOIL-1 helix-RBR (C460A) in the absence or presence of different di-Ub linkages. **b** ITC of isopeptide linked UbcH7(C86K)-Ub conjugate titrated into RNF216 RBR-helix (C688A, pSer719) in the absence or presence of different di-Ub linkages. **c** Summary of ITC experiment statistics. The titration of UbcH7-Ub into HOIL-1/M1 di-Ub was performed in duplicate, and statistics for both individual experiments are provided.

formation of RBR/UbcH7(C86K)-Ub/di-Ub complexes for both RNF216 RBR-helix (C688A, pSer719) and HOIL-1 helix-RBR (C460A), as shown by changes in elution volume and molecular mass (Supplementary Fig. 2a–c). The SEC-MALS derived molecular mass calculations support formation of 1:1:1 ternary complexes of the HOIL-1 or RNF216 RBR modules with a UbcH7(C86K)-Ub conjugate and allosteric di-Ub, although a ~20–25% discrepancy between the calculated and observed molecular weight of the ternary complexes suggests UbcH7(C86K)-Ub binding may be relatively weak under these conditions.

To obtain molecular insights into how allosteric di-Ub regulates HOIL-1 and RNF216 and how these RBRs bind the E2-Ub conjugate, we crystallised HOIL-1 and RNF216 in complex with UbcH7-Ub and allosteric Ub. To capture the stable transthiolation complexes, we used

active site mutant RBRs (HOIL-1 helix-RBR C460A or RNF216 RBR-helix C688A, pSer719) and the UbcH7(C86K)-Ub isopeptide conjugate. We copurified the HOIL-1(C460A)/UbcH7(C86K)-Ub/M1-linked di-Ub complex for crystallisation via size-exclusion chromatography (Supplementary Fig. 2d). We were unable to crystallise the RNF216(C688A, pSer719)/UbcH7(C86K)-Ub complex with K63 di-Ub and instead directly mixed RNF216 RBR-helix (C688A, pSer719) with a 1.5-fold molar excess of both UbcH7(C86K)-Ub and mono-Ub for crystallisation without further purification.

Crystallisation trials yielded crystals for the HOIL-1 and RNF216 complexes that diffracted to 3.08 Å and 3.03 Å, respectively. Molecular replacement using individual domains of previously solved structures and AlphaFold[29,30] models (see 'Methods' for details) allowed us to

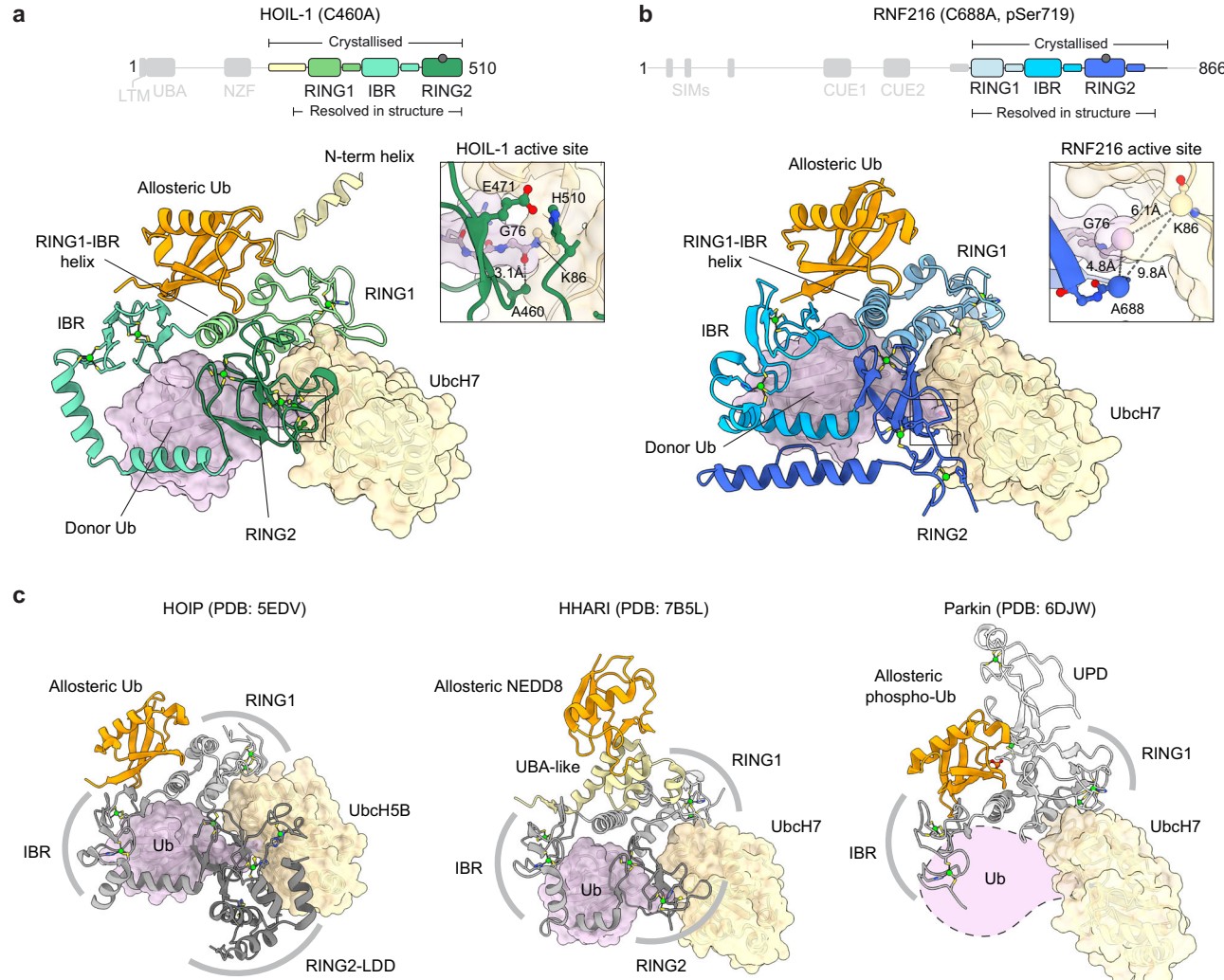

**Fig. 3 | Crystal structures of the HOIL-1 and RNF216 transthiolation complexes.**
**a** Overall structure of the HOIL-1 helix-RBR (C460A)/UbcH7(C86K)-Ub/Ub trans-thiolation complex (3.08 Å). HOIL-1 is shown in yellow and three shades of green, UbcH7 in tan, donor Ub in pink, allosteric Ub in orange. Only one of the two complexes present within the asymmetric unit is shown for simplicity. Supplementary Fig. 3 shows the full asymmetric unit and overlay of both complexes. **b** Overall structure of the RNF216 RBR-helix (C688A, pSer719)/UbcH7(C86K)-Ub/ Ub transthiolation complex (3.03 Å). RNF216 is shown in three shades of blue, UbcH7 in tan, donor Ub in pink, allosteric Ub in orange. The asymmetric unit

contains a single complex. The UbcH7(C86K)-Ub isopeptide bond is poorly resolved, but the close proximity of the backbone carbon of Ub G76 and the Cα atoms of UbcH7 C86K and RNF216 active site C688A indicate this complex repre-sents the relevant transthiolation complex (see inset). **c** Comparison of the struc-tures in panels (**a**) and (**b**) with published RBR transthiolation complexes HOIP (PDB: 5EDV[6]), HHARI (PDB: 7B5L[14]) and Parkin (PDB: 6DJW[12]). The Parkin complex was crystallised in the absence of Ub (approximate position represented by dotted line), and the RING2 domain is not resolved in the crystal structure. LDD linear Ub chain determining domain, UPD unique Parkin domain.

determine both complex structures (Fig. 3a, b, Table 1). Despite the comparable resolution, the electron density and model of the HOIL-1 complex are of considerably higher quality than those of the RNF216 complex structure (Table 1, Supplementary Fig. 3a, b). The latter fea-tures gaps in electron density and relatively weak density and high B-factors especially for the UbcH7 molecule (Supplementary Fig. 3b, c), allowing us to confidently place individual proteins and domains but providing limited information at the amino acid level, in particular the amino acid sidechains.

The HOIL-1 structure contains two near identical complexes (Cα RMSD 0.396 Å, Supplementary Fig. 3d) in the crystallographic asym-metric unit. Both complexes consist of the HOIL-1 RBR and the UbcH7-Ub conjugate but show clear electron density for only an additional mono-Ub, rather than the M1-linked di-Ub (Supplementary Fig. 3a). SDS-PAGE analysis of isolated crystals shows the presence of a mixture of free mono- and di-Ub, potentially due to di-Ub degradation during crystallisation (Supplementary Fig. 3e). The primary difference between the two complexes present in the asymmetric unit is presence

of density for parts of the N-terminal helix in the first complex, which is absent in the second complex (Supplementary Fig. 3a, insets). We therefore focus our analysis on the better resolved complex containing the N-terminal helix.

The RNF216 asymmetric unit comprises a single complex of the RNF216 RBR-helix, UbcH7-Ub and the allosteric mono-Ub (Supple-mentary Fig. 3b).

Both RBR/E2-Ub/Ub structures adopt a similar conformation that represents the active E2-Ub/E3 transthiolation complex, with the E2 and E3 active sites aligned for Ub transfer (Fig. 3a, b, insets). The RBR domains wrap around the Ub of the E2-Ub conjugate (donor Ub), stabilising the open E2-Ub conformation. The RBR-RING1 binds the E2 via the canonical RING1/E2 interface[6,14,31–34]. The E2-Ub isopeptide bond is placed into the RING2 active site poised for Ub transfer from the E2 to the E3. While the active site details are clearly resolved for the HOIL-1 complex (Fig. 3a, inset), weak density for the RNF216 complex means the isopeptide linkage between UbcH7 and Ub is not resolved. Nonetheless, the inferred proximity of the backbone

**Table 1 | Data collection and refinement statistics**

| | HOIL-1 helix-RBR (C460A)/UbcH7(C86K)-Ub/M1 di-Ub | RNF216 RBR-helix (C688A, pSer719)/UbcH7(C86K)-Ub/mono-Ub |
|---|---|---|
| **Data collection** | | |
| Space group | P2₁ | C2 |
| Cell dimensions | | |
| *a, b, c* (Å) | 83.95, 95.48, 115.00 | 152.55, 70.36, 65.15 |
| α, β, γ (°) | 90.00, 110.58, 90.00 | 90.00, 108.70, 90.00 |
| Resolution (Å) | 47.74–3.08 (3.25–3.08)ᵃ | 47.64–3.03 (3.21–3.03) |
| $R_{sym}$ or $R_{merge}$ | 0.230 (2.624) | 0.111 (0.979) |
| $I/\sigma I$ | 4.3 (0.9) | 5.6 (0.9) |
| Completeness (%) | 99.5 (96.9) | 98.4 (95.9) |
| Redundancy | 3.5 (3.6) | 3.4 (3.5) |
| **Refinement** | | |
| Resolution (Å) | 47.74–3.08 | 41.37–3.03 |
| No. reflections | 31,333 | 12,667 |
| $R_{work}/R_{free}$ | 0.2316/0.2664 | 0.3062/0.3389 |
| No. atoms | | |
| Protein | 8791 | 4401 |
| Ligand/ion | 14 | 22 |
| Water | 8 | 0 |
| *B*-factors | | |
| Protein | 110.9 | 92.5 |
| Ligand/ion | 92.9 | 86.7 |
| Water | 59.9 | – |
| R.m.s. deviations | | |
| Bond lengths (Å) | 0.002 | 0.002 |
| Bond angles (°) | 0.463 | 0.486 |

ᵃValues in parentheses are for highest-resolution shell.

carbon of Ub G76 and the Cα atoms of UbcH7 K86 (replacing the active site Cys) and RNF216 A688 (replacing the active site Cys) indicates this represents the relevant transthiolation complex (Fig. 3b, inset).

In both structures, the allosteric Ub binds the RING1-IBR helix and the IBR domain on the opposite side of the RBR in relation to the donor Ub (Fig. 3a, b). The observed 1:1:1 RBR:E2-Ub:Ub ternary complexes in the crystal structures are consistent with our in-solution SEC-MALS data (Supplementary Fig. 2a–c).

Overall, the HOIL-1 and RNF216 RBR/E2-Ub/Ub transthiolation complexes resemble previously determined active RBR/E2-Ub transfer complexes of HOIP[6] and HHARI[14], and a partly defined Parkin transthiolation complex lacking the RING2 and donor Ub[12] (Fig. 3c), but are different to E2-Ub bound inactive HHARI complexes that lack the allosteric activator[31,34].

**The RBR RING1 recruits the E2 via conserved interactions**
Our structures show that the RING1 domains of HOIL-1 and RNF216 interact in a canonical manner with the E2 UbcH7 as observed previously in structures of various E2-Ub conjugates bound to RING or RING1 domains (Supplementary Fig. 4a). Key interactions are mediated by UbcH7 F63 and the central helix of the RBR RING1 (Fig. 4a, b). Accordingly, HOIL-1 and RNF216 are completely inactive with UbcH7 F63A (Fig. 4c, d), and mutation of HOIL-1 V284 (V284A), one of the canonical RING/E2 interaction residues[3] that interacts with UbcH7 F63, abrogates E2-Ub discharge (Fig. 4e). The equivalent mutation in RNF216 (C517A) does not affect E2-Ub discharge activity, presumably due to the minimal difference between the short Cys sidechain and Ala, but mutation of the adjacent RNF216 Y519 reduces RNF216 E2-Ub discharge activity (Fig. 4f).

In contrast to canonical RING E3 ligases[33], but as previously observed for HOIP[6] and HHARI[14], the E2-Ub conjugate is stabilised in an open conformation that prevents direct Ub discharge and instead favours the transthiolation reaction to the RBR active site in the RING2 domain (Supplementary Fig. 4a).

Unlike canonical RING domains and all other RBRs, HOIL-1 features an extended loop between the seventh and eight zinc coordinating residues (RING1 Zn²⁺-loop II) that consists of eight residues in HOIL-1, but only 2-4 residues in all other RBRs and two residues in canonical RING domains (Fig. 4a, b, Supplementary Fig. 4b)[31,34]. The extended loop does not directly interact with UbcH7 in the HOIL-1 structure (Fig. 4a), and consistent with this, the HOIL-1 I326A mutant shows wild-type (WT) levels of activity (Fig. 4e). The comparison of the HOIL-1 RING1/E2 sub-complex with the structure of an active RNF4-RING E2-Ub complex[33] shows that the extended loop would prevent formation of the closed E2-Ub conjugate by clashing with Ub in that conformation (Supplementary Fig. 4c), therefore blocking Ub discharge from the E2 via the classic RING mechanism and forcing reaction via transthiolation[31,34]. Backbone atoms of RING1 Zn²⁺-loop II residues C323 and C332 in HOIL-1 and C559 in RNF216 form hydrogen bonds with the sidechain of UbcH7 K96, and the UbcH7 K96A mutation abrogates HOIL-1 and reduces RNF216 mediated E2-Ub discharge (Fig. 4c, d).

**RING2/E2-Ub interactions are mostly mediated by Ub**
At the other end of the RBR module, the RING2 domains of HOIL-1 and RNF216 interact with the E2-Ub conjugate (linked by an isopeptide bond in our structures), placing it into the RBR active sites, primed for Ub transthiolation. For both HOIL-1 and RNF216, the E2 engages in few interactions with the RING2 domains (Fig. 5a, b), suggesting that the RING1 domain is the main E2 recruiting module. This is in line with the large sequence and structural diversity in the RBR RING2 domains, especially in their C-terminal half. Instead, the donor Ub contributes most of the binding surface of the E2-Ub conjugate with the helix-RING2 module (Fig. 5a, b). The Ub C-terminus forms a parallel β-sheet with the second β-strand of the RING2 domain, and the I44 patch of the donor Ub binds the helix preceding the RING2 domain (IBR-RING2 helix) in both complexes (Fig. 5a, b). We previously showed that mutations in this helix and adjacent regions (R660A, E664A, I672A, I683A) disrupt RNF216 E2-Ub discharge and chain formation activity[23] and observe the same effect when mutating equivalent residues (L435A and M438A) in HOIL-1 (Fig. 5c).

**RBRs contain an allosteric Ub binding site in the RING1-IBR interface**
In both our RBR structures, a free Ub molecule binds to the allosteric site in the RING1-IBR interface. The I44 patch of the allosteric Ub interacts with the RING1-IBR helix, and the Ub C-terminus forms a β-sheet with the second β-strand of the IBR domain (Fig. 6a, b), analogous to how the helix-RING2 binds to donor Ub (Fig. 5a, b). In both cases, the RING1-IBR helix is in straight conformation, previously identified as a hallmark of the active RBR conformation[6,14,21]. The allosteric Ub binding sites in HOIL-1 and RNF216 are equivalent to those previously described for HOIP and Parkin, indicating a conserved allosteric activation mechanism of the RBR family (Fig. 3c).

HOIL-1 I358 is central to the allosteric Ub binding site and, accordingly, the HOIL-1 I358R mutant is less strongly activated by M1-linked di-Ub (EC₅₀ = 24 μM, Fig. 6c) compared to WT HOIL-1 (2 μM, Fig. 6c). Furthermore, ITC shows no binding of M1-linked di-Ub to HOIL-1 I358R, whereas weak ($K_D$ ~ 50 μM) but clearly detectable direct binding is observed for WT HOIL-1 (Supplementary Fig. 5a).

The HOIL-1 E383A mutation does not affect allosteric activation of HOIL-1 (Supplementary Fig. 5b). HOIL-1 E383 corresponds to HOIP E809, which interacts with two C-terminal Arg residues of the allosteric

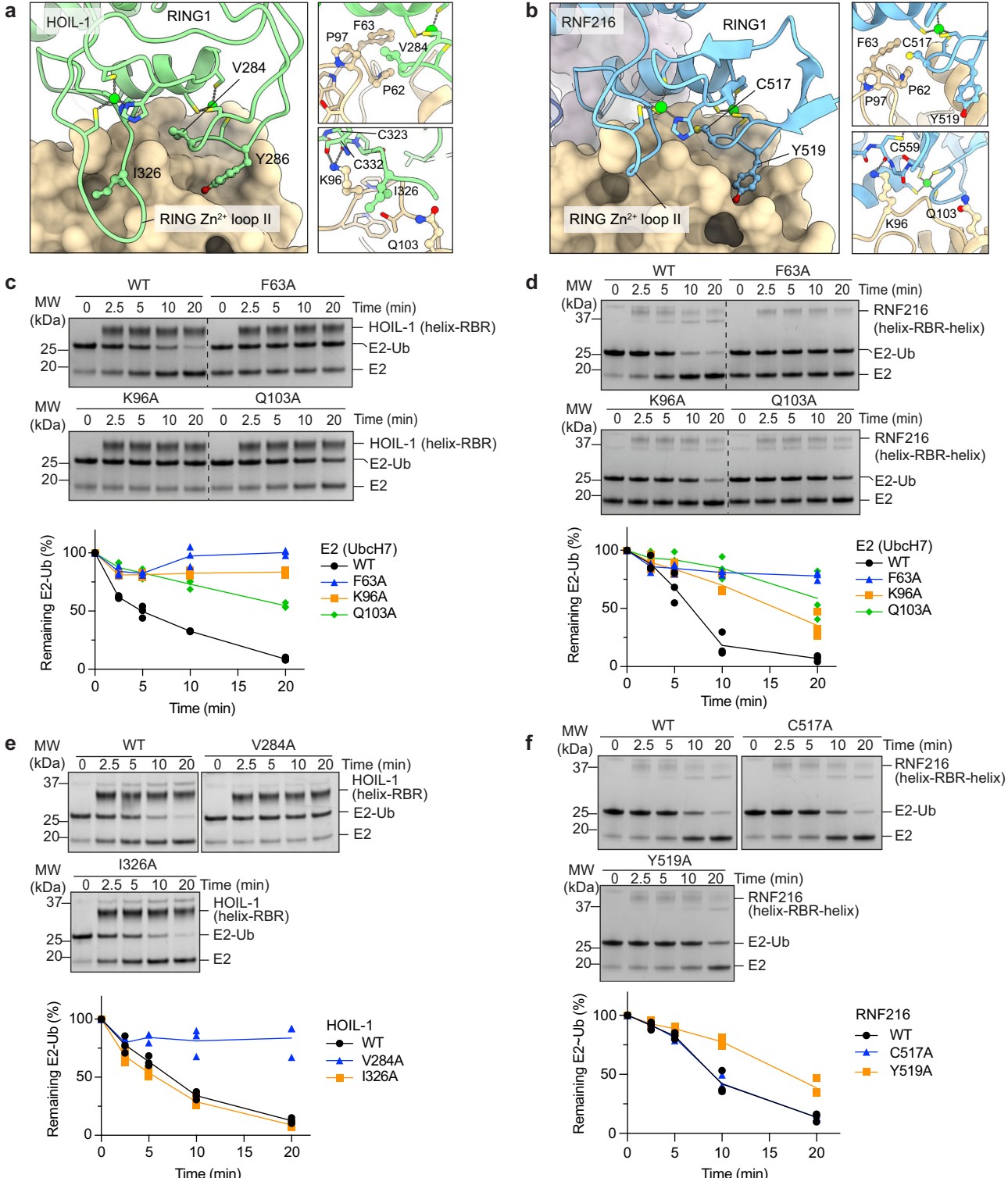

**Fig. 4 | Structural and functional analysis of the HOIL-1 and RNF216 RING1/E2 interfaces. a** HOIL-1 RING1/UbcH7 interface. The HOIL-1 RING1 domain (green) interacts with UbcH7 (tan surface model) via the central RING1 helix and adjacent loops. Insets show details of key interface residues presented in ball and stick depiction. **b** RNF216 RING1/UbcH7 interface. **c** Time-course of HOIL-1 catalysed E2-Ub discharge assay with UbcH7 mutants. Bottom panel shows quantification of three independent experiments. **d** Time-course of RNF216 catalysed E2-Ub discharge assay with UbcH7 mutants. Bottom panel shows quantification of three independent experiments. **e** Time-course of HOIL-1 catalysed E2-Ub discharge assay with HOIL-1 RING1 mutants. Bottom panel shows quantification of three independent experiments. **f** Time-course of RNF216 catalysed E2-Ub discharge assay with RNF216 RING1 mutants. Bottom panel shows quantification of three independent experiments. Source data are provided as a Source data file.

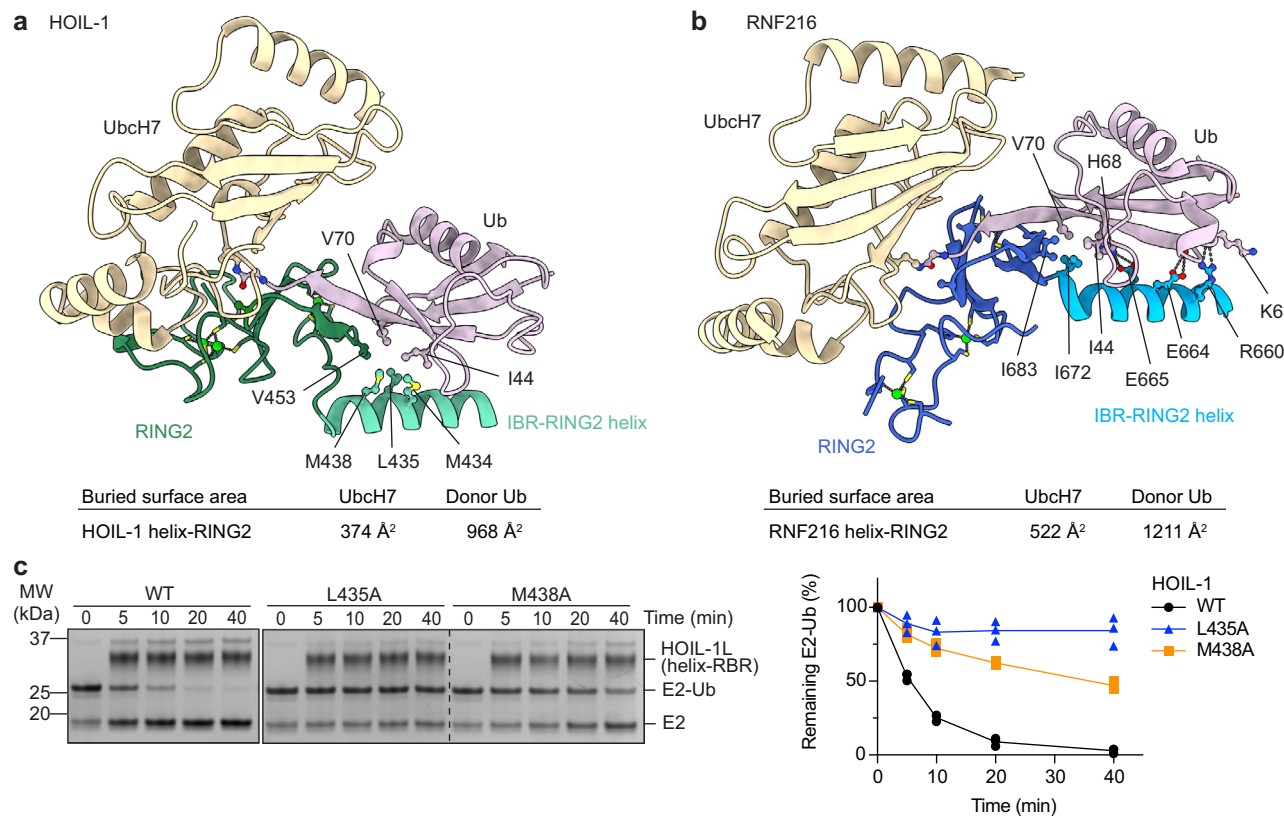

**Fig. 5 | Structural and functional analysis of the HOIL-1 and RNF216 RING2/E2-Ub interfaces. a** Binding of the UbcH7-Ub isopeptide conjugate to the HOIL-1 RING2 (dark green) and preceding helix (IBR-RING2 helix, light green) via the Ub I44 hydrophobic patch (pink). **b** Binding of the UbcH7-Ub isopeptide conjugate to the RNF216 RING2 (dark blue) and preceding helix (IBR-RING2 helix, light blue). Ub (pink) forms extensive interactions with polar and non-polar residues in the RING2 and IBR-RING2 helix. **c** Time-course of HOIL-1 catalysed E2-Ub discharge assay with WT HOIL-1 and IBR-RING2 mutants. The right panel shows quantification of three independent experiments. Source data are provided as a Source data file.

Ub and is critical for allosteric activation of HOIP by M1 di-Ub[6]. To test if the lack of effect in HOIL-1 is due to the presence of two other negatively charged residues (D384, D385) that may compensate for the loss of E383 (Fig. 6a), we compared allosteric activation of HOIL-1 E383A with HOIL-1 E383A/D384A/D385A. HOIL-1 E383A/D384A/D385A is still activated by M1-linked di-Ub (Supplementary Fig. 5c), suggesting that the interactions between the two Ub C-terminal Arg residues and HOIL-1 may not be as important for allosteric activation as is the case for HOIP.

Corresponding mutations in RING-IBR helix or IBR of RNF216 resulted in a significant decrease in baseline activity even in the absence of allosteric K63 di-Ub, preventing functional validation of the allosteric Ub binding site for RNF216.

In the HOIL-1 structure, despite the presence of di-Ub in the crystal (Supplementary Fig. 3e), we observe only a single allosteric Ub moiety. Given our earlier results that HOIL-1 is specifically activated by M1- or K63-linked di-Ub we aimed to further validate the Ub binding observed in our crystal structures, by testing if mono-Ub can activate HOIL-1 at high concentrations. In the presence of 200 µM mono-Ub, HOIL-1 is 2-times more active than in the absence of allosteric activator, but only ~20% as active as in the presence of 40 times less (5 µM) M1-linked di-Ub (Supplementary Fig. 5d), highlighting that mono-Ub can activate HOIL-1, albeit at a much weaker potency compared to M1-linked di-Ub.

To further deduce which Ub protomer of the di-Ub binds to the HOIL-1 canonical allosteric site defined in the crystal structures, we performed E2-Ub discharge assays in the presence of WT M1-linked di-Ub or M1-linked di-Ub where I44 was mutated to Ala in either the proximal or distal Ub protomer, or both protomers. These assays show

that a functional I44 patch in both Ub protomers is required for efficient allosteric activation of HOIL-1 (Supplementary Fig. 5e). Mutation of I44 in the proximal Ub (i.e., with a free C-terminus) has a stronger effect, suggesting that the proximal Ub binds to the canonical allosteric Ub binding site observed in the structure. Based on the arrangement in our crystal structure, the distal Ub may bind a helix N-terminal of the HOIL-1 RING1 domain, which is immediately adjacent to the allosteric Ub binding site, and position of the M1 and K63 residues in the proximal Ub agrees with M1- or K63 di-Ub binding to these sites (Fig. 6a). This helix is present in the HOIL-1 233-510 (helix-RBR) construct used in all our biochemical and biophysical assays. In the absence of M1-linked di-Ub, the HOIL-1 280-510 (RBR) construct lacking this helix has E2-Ub discharge activity comparable to the HOIL-1 helix-RBR construct (Fig. 6d, left panels), indicating that the helix is not required for general catalysis. However, HOIL-1 RBR lacking the N-terminal helix is only weakly activated by M1-linked di-Ub (Fig. 6d, right panels), suggesting that this helix contains a cryptic Ub binding site that is occupied by the distal Ub protomer of the allosteric M1-linked di-Ub and helps recruit allosteric di-Ub.

Like HOIL-1, RNF216 features a helix N-terminal to the RING1 domain (Fig. 1a) that in RNF216 contains a predicted Ub binding motif (motif interacting with ubiquitin; MIU)[35]. However, in contrast to HOIL-1, we find that the N-terminal helix does not contribute to allosteric activation of RNF216 by K63 di-Ub (Fig. 6e). Furthermore, mutation of the I44 patch in the distal rather than the proximal Ub has a larger effect on RNF216 allosteric activation (Supplementary Fig. 5f), suggesting that while the core allosteric binding site in the RING1-IBR helix is conserved, different RBRs utilise distinct mechanisms to implement chain-type specificity of the allosteric activation.

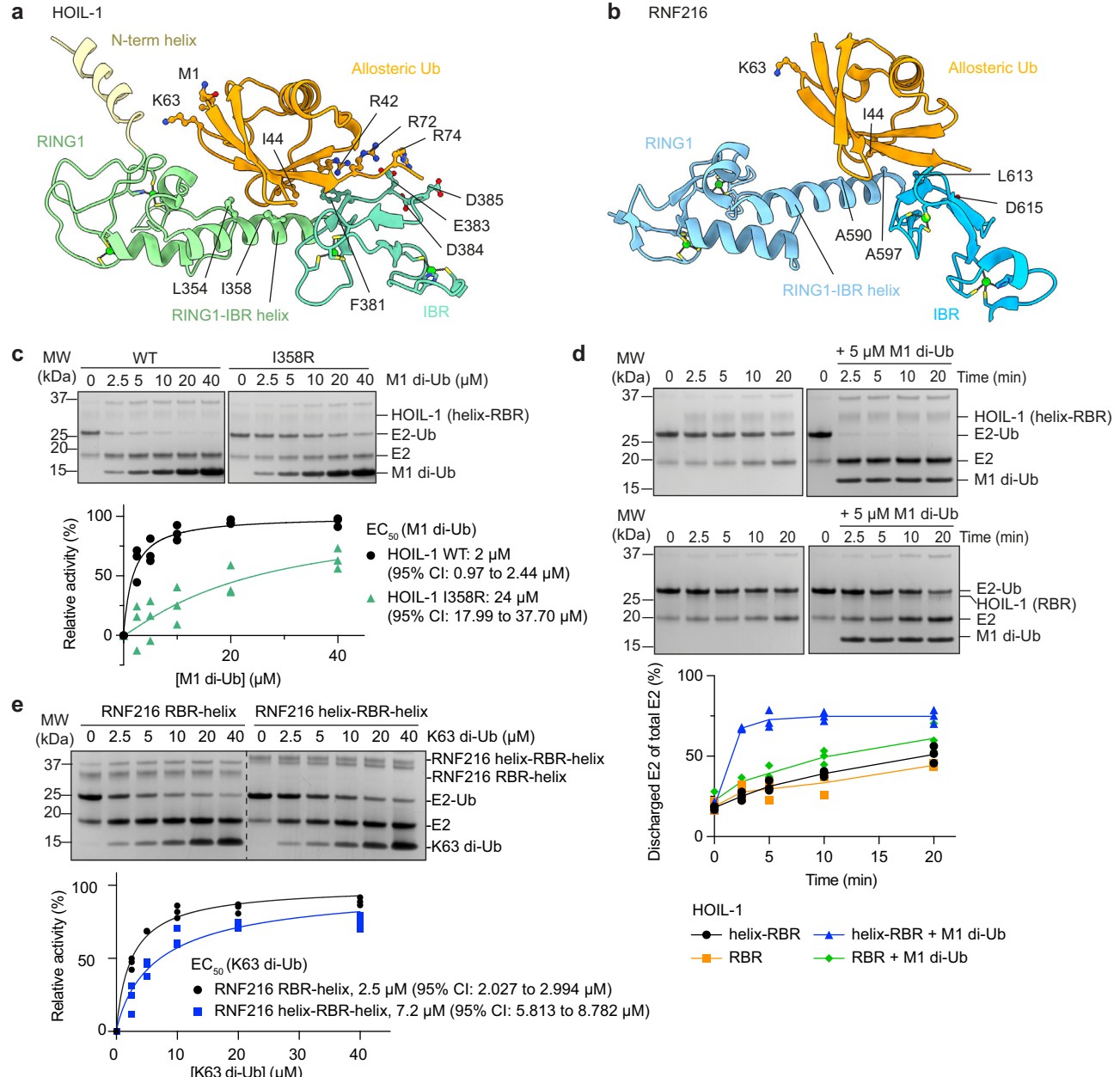

**Fig. 6 | Analysis of the allosteric Ub binding sites in HOIL-1 and RNF216. a** HOIL-1 allosteric Ub binding site. The Ub I44 hydrophobic patch binds to a conserved allosteric site formed by the HOIL-1 RING1-IBR helix and IBR. Key interacting residues and M1 and K63 that would form an (iso-)peptide bond with the distal Ub in di-Ub are shown in ball and stick depiction. **b** RNF216 allosteric Ub binding site. The Ub I44 hydrophobic patch binds to a conserved allosteric site formed by the RNF216 RING1-IBR helix and IBR. Key interacting residues and K63 are shown in ball and stick depiction. **c** WT and I358R HOIL-1 catalysed UbcH7-Ub discharge assay in the presence of increasing concentrations of M1-linked di-Ub. The bottom panel shows quantification of three independent experiments for EC$_{50}$ determination.

Nonlinear regression curves were fitted using the [agonist] vs. normalised responses−variable slope model. **d** Time-course of HOIL-1 catalysed E2-Ub discharge assay using constructs with (helix-RBR) or without (RBR) the N-terminal helix in the presence or absence of M1-linked di-Ub. The graph shows quantification of three independent experiments. **e** RNF216 catalysed UbcH7-Ub discharge assay using constructs with (helix-RBR-helix) or without (RBR-helix) the N-terminal helix, in the presence of increasing concentrations of K63-linked di-Ub. The bottom panel shows quantification of three independent experiments for EC$_{50}$ determination. Nonlinear regression curves were fitted using the [agonist] vs. normalised responses−variable slope model. Source data are provided as a Source data file.

## HOIL-1 features an unusual RING2 domain with a Zn2/Cys6 binuclear cluster

While the RING1 and IBR domains of the various RBRs are structurally conserved, the RING2 domains show larger variability, especially in the C-terminal region. For HOIL-1, previous sequence analysis showed that only the N-terminal half of the RING2 domain aligns well with other RING2 sequences (Fig. 7a)[8]. The C-terminal half, which normally contains four Cys/His zinc-coordinating residues, contains eight potential zinc coordinating residues (6 Cys, 2 His; Fig. 7a),

suggesting that the HOIL-1 RING2 may feature an additional zinc finger.

In our structure, only the N-terminal HOIL-1 RING2 portion (up to residue V475) follows the canonical RING2 fold, including the first RING2 zinc finger and the first zinc-coordinating Cys of the second RING2 zinc finger (C473), whereas the C-terminal portion of the RING2 adopts an unexpected fold (Fig. 7b). Instead of a second C-C-(H/C)-(C/H) zinc finger as in other RBRs, HOIL-1 contains a unique Zn2/Cys6 binuclear cluster, where 6 Cys residues coordinate two zinc ions in a C-

x19-C-x-C-x6-C-x3-C-x2-C configuration. Each zinc ion is tetrahedrally coordinated and shares two Cys ligands with the other zinc ion (Fig. 7b–d, Supplementary Fig. 6a). Based on sequence and structure homology searches, this fold is not found in any other mammalian zinc coordinating domain. Other proteins that contain Zn2/Cys6 binuclear clusters are fungal transcription factors like GAL4 and HAP1, which coordinate the two zinc atoms in a very similar manner, sharing two Cys ligands between the two zinc ions[36–38], but otherwise do not show any structural or functional resemblance to the HOIL-1 RING2 (Supplementary Fig. 6).

The activity and substrate specificity of HOIL-1 have been an enigma for many years[39]. In recent years, HOIL-1 was characterised as a Ser/Thr-specific E3[25–27], but it has more recently been suggested that HOIL-1 has a surprising, non-proteinaceous activity and can ubiquitinate oligo- and polysaccharides such as maltose and glycogen[24]. Previous work on the RBRs HOIP and RNF216 showed that zinc finger insertions in the C-terminal portion of the RBR RING2 domains are required for acceptor Ub binding and define substrate or chain type specificity[23,40], suggesting that the C-terminal Zn2/Cys6 binuclear cluster in HOIL-1 may be an adaptation to accommodate non-proteinaceous substrates. To test this hypothesis, we compared the maltose ubiquitination activity of HOIL-1 to the activity of other RBRs (RNF216 and HOIP) and the HECT E3 ligase HECTD1. Consistent with previous reports, HOIL-1 ubiquitinates maltose, indicated by altered Ub migration in SDS-PAGE and confirmed by mass spectrometry (Supplementary Fig. 7a, b). Unexpectedly, we find that RNF216, which normally forms K63-linked polyubiquitin chains[23,35,41], can also ubiquitinate maltose to a similar degree as HOIL-1 (Supplementary Fig. 7a, c). HOIP is one of the most chain-type specific E3 ligases[17,40,42] and with native Ub efficiently forms M1-linked Ub chains. However, with N-terminally blocked His-Ub, which cannot form M1-linked chains, we observe that HOIP can also ubiquitinate maltose, albeit with much lower activity than HOIL-1 or RNF216 (Supplementary Fig. 7a). Maltose is not ubiquitinated in the absence of an E3 nor by HECTD1, showing that the activity is catalysed by specific E3 ligases in vitro.

The activity of many RBRs is dependent on a Cys, His, Glu/Gln catalytic triad[6,8,40,43]. The catalytic Cys is essential for RBR function, but the importance of the other residues, which are not fully conserved[8], varies between different family members. In HOIL-1, the residue aligning with the catalytic His of other RBRs is W462 (Fig. 7a) suggesting a distinct active site arrangement. In our structure, the side chain of the very C-terminal residue, H510, is not involved in $Zn^{2+}$ coordination, but points towards the HOIL-1 active site and may form a catalytic triad with C460 and E471 (Fig. 7a, b). To test this hypothesis, we compared the activity of the HOIL-1 H510A mutant with WT HOIL-1 and the HOIL-1 C460A active site mutant. As expected, the C460A mutant shows no activity in either the E2-Ub discharge assay (Fig. 7e) or the maltose ubiquitination assay (Fig. 7f). The H510A mutant can still efficiently discharge Ub from the E2, showing that H510 is not required for Ub transthiolation (Fig. 7e). However, we observe appearance of a slower migrating band that decreases in intensity under reducing conditions (Fig. 7e, red arrowhead), suggesting that the HOIL-1 H510A mutant stabilises the HOIL-1-Ub thioester. We hypothesise that H510 is critical for Ub transfer from the HOIL-1 active site to the substrate. In agreement with this hypothesis, we observe that HOIL-1 H510A cannot efficiently ubiquitinate maltose (Fig. 7f), likely because H510 is required for deprotonating the incoming nucleophile, as has been shown for the canonical catalytic triad residue H887 in HOIP[40].

## Discussion

Since the realisation that RBR E3 ligases utilise a RING/HECT hybrid mechanism a decade ago[5], research on three (HOIP, Parkin, HHARI) of the 14 family members started to unravel the RBR mechanism and its

regulation[8]. However, limited work on other family members, left the question of which features define the RBR family as a whole, and which are protein-specific adaptations, unanswered.

Our work on the RBR E3 ligases HOIL-1 and RNF216 presented here, together with previous work on HOIP[6,17,39,40,42,44], Parkin[5,10–12,21,22,43,45–48] and HHARI[5,13,14,16,31,34,49], defines a conserved RBR/E2 transthiolation complex conformation that enables efficient Ub transfer from the E2 to the RBR active site. This common RBR transthiolation complex is defined by: (1) the conserved E2 binding site in the RING1 domain; (2) stabilisation of the open E2-Ub conformation, allowing (3) alignment of the E2 and E3 active sites; (4) an allosteric Ub/UBL binding site in the RING1-IBR interface, for (5) allosteric activation by distinct Ub or UBL molecules; and (6) a conserved donor Ub binding site in the helix-RING2 module (Fig. 8). It also becomes clear that RBRs are generally activated by the Ub linkage that is the product of their catalytic reaction (HOIP[6], RNF216[23]) or a Ub species closely associated with their signalling function (Parkin[11], HOIL-1[24], and this study), suggesting that RBRs are feedforward enzymes that accelerate their signalling functions (Fig. 8).

However, exceptions to these rules exist, representing adaptations of individual RBRs for specific functions, modes of regulation, and interactions with other proteins such as substrates, cofactors or specific E2s. For example, while Parkin, HOIP, HOIL-1 and RNF216 are directly activated by Ub species binding to the allosteric site, HHARI is activated by the UBL NEDD8, but only when NEDD8 is attached to a cullin scaffold protein[16]. NEDD8 also does not directly bind to the allosteric site in HHARI, but to the N-terminal UBA-like domain, which in turn occupies the conserved allosteric site[14]. Nevertheless, structural data show that this binding induces the same conformational changes, i.e., straightening of the RING1-IBR helix, as observed in other RBRs[6,14,21], suggesting that these diverse mechanisms ultimately converge upon a common activating mechanism. A recent study proposes that HHARI has additional cullin-independent functions[15], and it will be interesting to investigate if and how HHARI is allosterically activated under these settings.

Our data show that the allosteric activation enhances the first RBR reaction step (E2-to-RBR transthiolation) but not the second step, i.e., Ub transfer from the E3 to the substrate. This is supported by ITC data showing that allosteric activation enhances E2-Ub recruitment, presumably by inducing the straightening of the RING1-IBR helix and stabilising the correct conformation of the RBR module for efficient E2-Ub conjugate binding. The specific interaction between donor Ub and the RBR may be less important for the second catalytic step when the donor Ub C-terminus is covalently loaded onto the RBR active site through a thioester bond and therefore locked in the correct position to react with the substrate nucleophile. Comparison of RNF216 structures representing different stages of the catalytic cycle highlight the conformational flexibility of the RBR module (Supplementary Fig. 8a).

While HOIL-1 and RNF216 are activated by specific di-Ub linkages, our structures only show binding of a single Ub to the canonical allosteric binding site in the RING1-IBR interface. Our work shows that, at least for HOIL-1, high concentrations of mono-Ub can weakly activate E3 ligase activity, supporting the relevance of our structures. In combination, our experiments suggest that the allosteric Ub visualised in our structures is responsible for inducing the conformational changes required for allosteric activation, whereas the role of the 2nd Ub protomer in the di-Ub may be to increase avidity and enable chain-type specificity. With our structures, we cannot delineate how the RBRs distinguish between different poly-Ub linkages at the allosteric site, but our biochemical data with I44A mutants in either the proximal or distal Ub protomer of the di-Ub suggest that HOIL-1 and RNF216 may have adapted different ways of recognising allosteric di-Ub. For HOIL-1, mutation of the proximal Ub shows the larger effect whereas the distal Ub appears to be more important for allosteric activation of RNF216 by K63 di-Ub. Furthermore, in HOIL-1, the helix N-terminal to the RING1

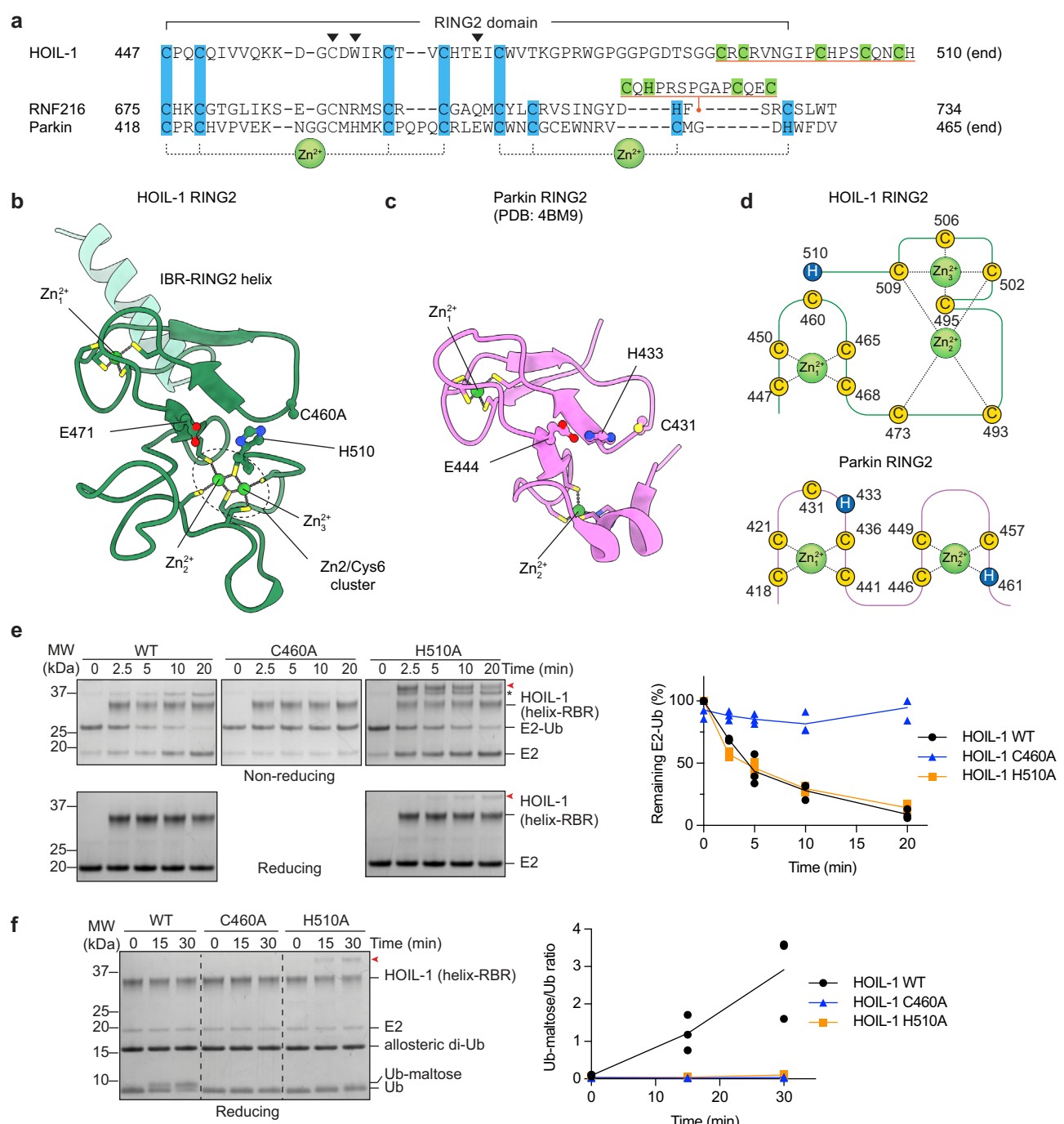

**Fig. 7 | The HOIL-1 RING2 domain contains a Zn2/Cys6 binuclear cluster.**
**a** Sequence alignment of RING2 domains from HOIL-1, RNF216 and Parkin. Zinc binding residues are highlighted blue and green. Position of the catalytic triad residues are indicated by black arrowheads. The RING2 domains of HOIL-1 and RNF216 contain distinctive zinc binding extensions or insertions (underlined in orange). Conversely, Parkin has a canonical RING2 domain, with two zinc-binding motifs and a conserved catalytic triad of Cys, His, Glu. **b** Structure of the HOIL-1 RING2 domain showing the Zn2/Cys6 binuclear cluster and active site residues (with active site Cys460 mutated to Ala). **c** Structure of the Parkin RING2 domain (PDB: 4BM9)[43] with canonical zinc binding arrangement and active site residues. **d** Schematics showing arrangement of HOIL-1 and Parkin RING2 zinc-coordinating residues. **e** Time-course of HOIL-1 catalysed E2-Ub discharge assay with HOIL-1 RING2 mutants. The red arrowhead indicates the accumulation of the E3-Ub thioester intermediate that is reduced by addition of DTT shown in the lower panels, as well as a small amount of HOIL-1 autoubiquitination product that is stable under reducing conditions. The asterisk indicates E2 disulfide dimer that is also present in the WT reaction and reduced by addition of DTT shown in the lower panels. The right panel shows quantification of three independent experiments. **f** Time-course of maltose ubiquitination by HOIL-1 WT and mutants. The red arrowhead indicates autoubiquitination of HOIL-1 H510A. The right panel shows quantification of three independent experiments. Source data are provided as a Source data file.

domain is critical for allosteric activation, while a similar helix in RNF216 is not required for allosteric activation. These observations again exemplify that while the general mechanism between RBRs is conserved (allosteric activation by Ub binding to the RING1-IBR

interface), individual RBRs have adapted in distinct ways to finetune the mechanism of activation.

Most of the interactions of the E2-Ub conjugate with the RBR IBR-RING2 module in our structures stem from interactions of Ub with the

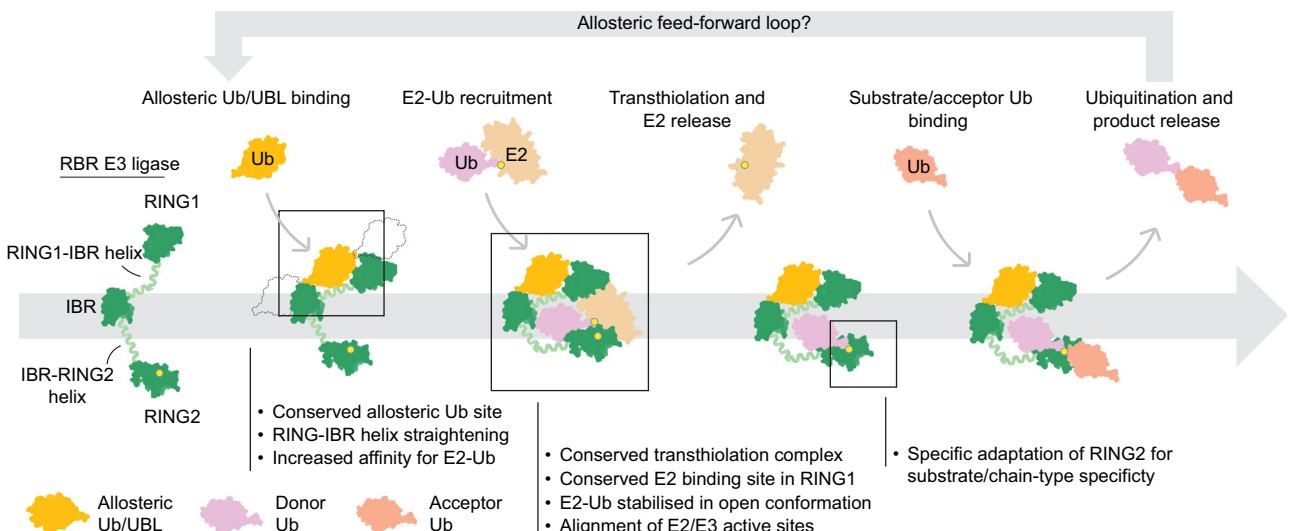

**Fig. 8 | Working model of the RBR mechanism.** RBR E3 ligases (green) are activated by allosteric Ub or UBLs which bind a conserved site in the RING1-IBR helix. Allosteric Ub or UBL binding at this site induces a conformational change that promotes E2-Ub binding. E2-Ub binds to the RBR via a canonical E2 interface in the RING1 domain and extensive interactions between donor Ub and the IBR-RING2 helix, establishing a conserved transthiolation complex in which E2-Ub is stabilised in the open conformation and E2/E3 active sites (yellow circles) are aligned. Ub transfer from E2 to the RING2 active site (transthiolation) permits release of E2, and binding of incoming substrate molecules (including acceptor Ub) to the RING2 domain. Structural adaptations within the RING2 domain confer substrate and Ub chain type specificity to the RBR E3 ligases. Ub transfer to the bound substrate and subsequent release of the ubiquitinated substrate completes the cycle. The poly-Ub product of catalysis may function as an allosteric activator itself, providing feed-forward amplification of the ubiquitination cycle. The model combines this work and previous studies on HOIP, HHARI, Parkin and RNF216 as detailed in the main text.

IBR-RING2 helix and the N-terminal portion of the RING2. These interactions are conserved in other RBR/E2-Ub complexes of HOIP, HHARI and Parkin[6,7,14,15,23,45]. This may be a specific adaptation that allows stable formation of the E2-Ub/E3 transthiolation complex but enables release of the E2 once Ub is transferred to the E3 active site. Comparison of the RNF216/E2-Ub/Ub complex structure with our previous structure of product bound RNF216 RING2 shows that the E2 and acceptor Ub binding sites overlap (Supplementary Fig. 8b), which previously has been observed for HOIP[6,40], and underscores that E2 release is critical for subsequent acceptor Ub binding.

The HOIL-1 RING2 domain features an unusual Zn2/Cys6 binuclear cluster where six Cys residues coordinate two zinc ions. Our data show that this binuclear cluster including the very C-terminal H510 is an integral part of the HOIL-1 active site. It was previously observed that zinc finger insertions in the HOIP and RNF216 RING2 domains are substrate or acceptor Ub recruitment sites[8,23,40], and we speculated that the HOIL-1 Zn2/Cys6 binuclear cluster would perform a similar function. Given the recent observations that HOIL-1 can directly ubiquitinate glucosaccharides such as maltose and glycogen[24], we reasoned that the HOIL-1 Zn2/Cys6 binuclear cluster may be a specific saccharide substrate binding platform. We confirmed that HOIL-1 can ubiquitinate maltose in vitro, however, we also observed that the RBRs RNF216 and HOIP that have been shown to catalyse K63- and M1-linked polyubiquitination, respectively, can also ubiquitinate maltose under the same reaction conditions. RNF216 and HOIP feature very different RING2 structures compared to HOIL-1 and to each other, underscoring that the HOIL-1 Zn2/Cys6 binuclear cluster may not be a specific adaptation to allow maltose ubiquitination, but may still play important roles to recruit physiological substrates. Further research in more physiological settings is required to confirm whether saccharides are genuine HOIL-1 biological substrates and to unravel the function of HOIL-1's Zn2/Cys6 binuclear cluster.

## Methods
### Plasmids
The DNA sequences coding for full-length human HOIL-1 and mouse HECTD1 (amino acids 2106–2618), both optimized for expression in E. coli (gBlock Gene Fragment, IDT Singapore), were inserted into the pOPINB vector (Berrow et al. [50]) using in-fusion cloning (Takara Bio). UBE1, RNF216, HOIP, UbcH7, native Ub and M1 di-Ub constructs were described previously[6,23]. Shorter RBR constructs were subcloned from the full-length plasmids, and mutations were introduced using the QuikChange II (Agilent) or the Q5 (NEB Biolabs) site-directed mutagenesis kits. The non-cleavable His-Ub construct was derived from a previously described His-Ub construct[6] with the human rhinovirus (HRV) 3C protease site removed by Q5 site-directed mutagenesis. Coding sequences for all plasmids were verified using Sanger sequencing (AGRF, Melbourne). Plasmids for 6xHis-tagged full-length human HOIL-1 and HOIL-1 233–510 codon-optimised for expression in E. coli have been deposited to Addgene (www.addgene.org) with accession codes 193858 and 193859. Oligonucleotide sequences are listed in Supplementary Dataset 1.

### Protein expression and purification
Heat-shock transformed E. coli BL21(DE3) were grown in 2x YT media with antibiotics until OD$_{600}$ -0.8, induced for expression with 0.5 mM IPTG (Gold Biotechnology) and grown overnight at 20 °C before harvesting by centrifugation. Cell pellets were kept at −20 °C and used for purification within 6 months. For HOIL-1, RNF216 and HOIP expression, cultures were supplemented with 0.5 mM ZnCl$_2$ before induction.

For purification, cell pellets were lysed by addition of lysozyme and sonication in the presence of protease inhibitors (PMSF, leupeptin), DNase I and MgCl$_2$. Cell lysates were clarified by centrifugation.

To purify His-tagged HOIL-1, cleared cell lysates were captured by Ni-NTA His-bind resin (Merk Millipore) equilibrated in high salt buffer (50 mM Tris pH8, 500 mM NaCl, 10% glycerol (v/v) and 10% sucrose (w/v)) in gravity columns. Bound resin was washed with high salt buffer before protein was eluted in high salt buffer supplemented with 300 mM imidazole. Eluted protein was subjected to His-HRV 3C protease cleavage in dialysis tubing and dialysed against high salt buffer at 4 °C overnight. Cleaved HOIL-1 protein was passed through Ni-NTA His-bind resin to remove His-HRV 3C protease and cleaved His-tag. HOIL-1 was further purified on a Superdex 200 Increase 10/300 GL (Cytiva) column equilibrated in buffer containing 10 mM HEPES pH7.9

and 100 mM NaCl on an ÄKTA pure 25 FPLC system (Cytiva) controlled by Unicorn 7.5 software (Cytiva). HOIL-1 protein used for maltose ubiquitination assay was exchanged into DPBS pH 7.9 buffer at this step.

HOIL-1 mutants were purified using similar methods, but without cleaving the His-tag.

RNF216, HOIP RBR (amino acids 696–1072), and HECTD1 were purified by Ni-NTA affinity chromatography and cleaved by HRV 3 C protease as described for HOIL-1. RNF216 used for crystallography, HOIP, and HECTD1 were additionally purified by anion-exchange chromatography. Ni-NTA purified protein was diluted with 50 mM Tris pH 8.0 to a final NaCl concentration of 50 mM, and applied onto a ResourceQ column (6 mL, Cytiva). The bound protein was washed with 10 column volumes (CV) of buffer containing 50 mM Tris pH 8.0 or 10 mM HEPES pH 7.9, 50 mM NaCl and eluted over a 10 CV gradient from 0–100% with a buffer containing 50 mM Tris pH 8.0 or 10 mM HEPES pH 7.9, 1 M NaCl. All RNF216 proteins, HOIP, and HECTD1 were further purified on a Superdex 200 Increase 10/300 GL (Cytiva) column equilibrated in buffer containing 10 mM HEPES pH 7.9 and 100 mM NaCl. WT and mutant RNF216 used for activity assays were purified as above, but without cleaving the His-tag.

UbcH7 was purified as described for HOIL-1.

Human E1 (UBE1) was purified using a modification of the GST-Ub capture protocol based on existing methods[51]. Ub and M1-linked di-Ub were purified as previously described[6]. Briefly, bacterial lysates were diluted in sodium acetate, pH 4.5 to precipitate non-Ub proteins. Lysates were cleared using centrifugation, and Ub was captured on a ResourceS column (6 ml, Cytiva) equilibrated in 50 mM sodium acetate, pH 4.5 and eluted in a 0–500 mM NaCl gradient. Ub was further purified via size-exclusion chromatography on a Superdex 75 16/600 pg (Cytiva) in buffer containing 10 mM HEPES pH 7.9 and 100 mM NaCl.

K63 and K48 linked di-Ub were assembled enzymatically following published methods[6,23,52]. In brief, a combination of Uev1a and Ubc13 E2 enzymes were used to assemble K63-linked Ub chains, while K48 was assembled using Cdc34. For both chain-types, di-Ub species were isolated using cation-exchange chromatography and further purified by size-exclusion chromatography as described above for Ub purification. Blocked K63 di-Ub and mutants thereof were generated as above but by substituting native Ub with an equimolar mixture of K63R Ub and Ub ΔGG (Ub 1-74) with or without additional I44A mutations. The remaining di-Ub species used in Figs. 1b and 1e were kindly provided by David Komander (WEHI)[52]. K27-linked di-Ub was obtained from UbiQ Bio, Netherlands.

All recombinant proteins were generally flash frozen in liquid nitrogen in small aliquots (100 μl or less) and stored at −80 °C until use.

## In vitro phosphorylation of RNF216

RNF216 used for crystallisation, ITC, and SEC-MALS was phosphorylated on Ser719 (pSer719) in vitro using recombinant ERK2, ref. [23]. Briefly, purified RNF216 RBR-helix (200 μM) was incubated with 20 μM human His-tagged, active ERK2 (pT185/pY187 ERK2) for 2–3 h at 30 °C in 50 mM Tris pH 7.5, 100 mM NaCl, 10 mM MgCl₂, 2 mM DTT and 10 mM ATP. His-ERK2 was subsequently removed by Ni-NTA agarose and phosphorylated RNF216 further purified by size-exclusion chromatography as above. Phosphorylation was confirmed by Phos-tag SDS-PAGE analysis (SuperSep Phos-tag Precast Gels, Fujifilm Wako Chemicals) and intact mass spectrometry.

## Stable UbcH7-Ub conjugate preparation

The stable UbcH7(C86K)-Ub conjugate was prepared following a published method to generate UbcH5A(C85K)-Ub[33]. Briefly, Purified UbcH7 (C86K) was mixed with UBE1 and Ub in a molar ratio 0.033 E1:1 E2:1.5 Ub and buffer exchanged into 50 mM Tris pH 10, 150 mM NaCl using a HiTrap desalting column (Cytiva). Buffer exchanged proteins

were supplemented with 10 mM MgCl₂, 1 mM TCEP and 5 mM ATP and incubated at 37 °C overnight to allow for UbcH7-Ub formation. UbcH7-Ub conjugate was purified by size-exclusion chromatography on a Superdex 75 10/300 GL column (Cytiva) equilibrated in 10 mM HEPES pH 7.9, 100 mM NaCl.

## Crystallisation

Crystallisation trials were performed using the sitting drop vapour diffusion method at the CSIRO Collaborative Crystallisation Centre (C3) and the Monash Macromolecular Crystallisation Facility (MMCF) using commercial screens.

**HOIL-1/UbcH7-Ub/M1 di-Ub complex.** HOIL-1 helix-RBR (residues 233-510, C460A), UbcH7 (C86K)-Ub conjugate and M1 di-Ub were mixed at a 1:1.5:1.5 molar ratio and purified using a Superdex 200 Increase 10/300 GL size-exclusion chromatography column (Cytiva) equilibrated in buffer containing 10 mM HEPES pH 7.9, 100 mM NaCl. Fractions containing all three proteins were pooled, concentrated to 10 mg/mL and used in crystallisation screens. Initial hits were obtained from the BCS PEG Smears screen (Molecular Dimensions) condition 28% PEGSB, 0.2 M NaCl, 0.1 M Na-phosphate pH 6.2. Hits were optimised by varying drop ratio and seeding. Final crystals were obtained by mixing 2.6 μl protein solution with 1 μl reservoir solution (28% PEGSB, 0.2 M NaCl, 0.1 M Na-phosphate pH 6.2). Crystals were cryo-preserved in reservoir solution supplemented with 15% v/v glycerol before cryo-cooling in liquid nitrogen.

**RNF216/UbcH7-Ub/Ub complex.** Purified RNF216 RBR-helix (residues 511-784, C688A, pSer719) was mixed with UbcH7 (C86K)-Ub conjugate and mono-Ub at a 1:1.5:1.5 molar ratio (final total protein concentration ~16.5 mg/mL) and 5 mM TCEP. Final crystals were obtained by mixing 150 nl protein solution with 300 nl reservoir solution (2 M ammonium sulfate) after >55 days at 25 °C. Crystals were cryo-preserved in reservoir solution supplemented with 20% v/v glycerol before cryo-cooling in liquid nitrogen.

## Diffraction data collection, processing, and structure solution

**General.** Crystal diffraction data were collected at the Australian Synchrotron MX2 beamline[53] at a wavelength of 0.9537 Å and 100 K temperature. Datasets were indexed and integrated with XDS[54] and merged with AIMLESS[55] using the automated processing pipeline at the Australian Synchrotron[53]. Phasing was performed by molecular replacement with Phaser[56] as detailed below. The covalent link between Ub G76 and UbcH7 K86 was defined using AceDRG[57] included in CCP4i2[58] and appropriate restraints included during refinement. Iterative cycles of model-building and refinement were completed using Coot[59] and Phenix.refine[60,61]. Structures were validated using MolProbity[62], CheckMyMetal[63] and the PDB Validation server[64]. Data processing and refinement statistics are shown in Table 1. All structural figures were prepared using UCSF ChimeraX[65]. Buried surface areas for protein-protein interactions were calculated with the measure buried area command in ChimeraX, using RING2 residues 425-510 for HOIL-1 and 655-772 for RNF216.

**HOIL-1/UbcH7-Ub/M1 di-Ub complex.** The complex structure was solved using molecular replacement in Phaser. Due to the flexibility of the HOIL-1 chain, initial molecular replacement searches included two molecules each of UbcH7 (PDB 4Q5E, chain C[66]) and Ub (PDB 4Q5E, chain B[66]). After confirming that Phaser identified realistic E2-Ub conjugates, in a second round of molecular replacement, two copies of the AlphaFold[29,30] model of the HOIL-1 RING1-IBR module (HOIL-1 residues 270–409) were included, followed by addition of two copies of the HOIL-1 RING2 domain (AlphaFold model, residues 442–510) and two more Ub molecules. Coot was used to rearrange the chains into a single compact asymmetric unit, and to manually build the N-terminal helix

(chain A) and IBR-RING2 helices (chains A and B). Data collection and refinement statistic are reported in Table 1. 94.13% of residues are in the favoured and 5.69% in the allowed region of the Ramachandran plot, and 0.18% are outliers.

**RNF216/UbcH7-Ub/Ub complex.** The complex structure was solved using multiple rounds of molecular replacement in Phaser. The initial search included one copy each of the RNF216 helix-RING2-helix/Ub complex (PDB 7M4O, ref. [23]), RNF216 RING1-helix (residues 511–598, modelled in AlphaFold) and free Ub (PDB 1UBQ, ref. [67]). In the second round of molecular replacement, we added UbcH7 (PDB 4Q5E, chain C[66]), followed by manual addition of the IBR (residues 600–649) from PDB 7M4M, ref. [23]. Connecting helices and linkers were built manually in Coot. ISOLDE[68] was used to model poorly refining regions. The electron density for UbcH7 is weak (Supplementary Fig. 3b), presumably due to high flexibility, reflected in high B-factors (Supplementary Fig. 3c) and relatively poor refinement statistics (Table 1). We therefore used a high-resolution UbcH7 structure (PDB 7V8F, ref. [69]) to generate dihedral angle reference model restraints during refinement in phenix.refine[70]. Data collection and refinement statistic are reported in Table 1. 93.87% of residues are in the favoured and 6.13% in the allowed region of the Ramachandran plot, with no outliers.

### Isothermal titration calorimetry (ITC)

Isothermal titration calorimetry (ITC) was conducted using an ITC200 calorimeter (Microcal). UbcH7(C86K)-Ub isopeptide conjugate (500 μM) was titrated into 50 μM RNF216 RBR-helix (511-784, C688A, pSer719) or HOIL-1 helix-RBR (233-510, C460A) with or without 150 μM di-Ub. The phosphorylated RNF216 construct was used here as this was previously shown to be most active in E2-Ub discharge assays[23]. All titrations were performed with $1 \times 1\,\mu l$ and $19 \times 2\,\mu l$ injections at 25 °C, with both components in 10 mM HEPES pH 7.9, 100 mM NaCl. Data were collected using ITC200 software (GE Healthcare) and processed and analysed using Origin software (Microcal).

### SEC-MALS

Size-exclusion chromatography coupled to multi-angle light scattering (SEC-MALS) was performed using a Superdex 75 Increase 10/300 GL column (Cytiva) coupled with a DAWN HELEOS II light scattering detector (Wyatt Technology, USA) and an Optilab T-rEX refractive index detector (Wyatt Technology, USA). The system was equilibrated in 10 mM HEPES (pH 7.9), 100 mM NaCl and calibrated with BSA (2 mg/mL). For each experiment, 100 μL of purified protein (1–2 mg/mL) was injected onto the column and eluted at 0.5 mL/min flow rate. The following proteins were used: RNF216 RBR-helix (511-784, C688A, pSer719) at 100 μM, HOIL-1 helix-RBR (233-510, C460A) at 100 μM, UbcH7(C86K)-Ub at 120 μM, K63 di-Ub at 120 μM, M1 di-Ub at 120 μM. All data were collected and processed using ASTRA software (v7.3.1.9, Wyatt Technology).

### UbcH7-Ub discharge assays

E2-Ub discharge experiments are based on published methods[33,71,72]. Unless otherwise stated, UbcH7 was charged with Ub by incubating 20 μM UbcH7 with 200 nM UBE1 and 50 μM Ub, 10 mM MgCl2 and 10 mM ATP in DPBS or 10 mM HEPES pH 7.9, 100 mM NaCl at 37 °C for 10 min. The reaction was depleted of ATP by incubating with 2 U apyrase (Sigma-Aldrich A6410) per 100 μl reaction at room temperature for 5 min. For time course E2-Ub discharge assays, the charged E2-Ub conjugate was mixed with E3 (either alone or pre-incubated with di-Ub species) for final concentrations of 1 μM E3, 10 μM E2-Ub and 5 μM di-Ub (where applicable). Reaction samples were taken at time points as indicated and quenched by mixing with 2x LDS buffer (Invitrogen) without reducing agent.

To perform HOIL-1 E2-Ub discharge for allosteric di-Ub EC50 calculation, E2-Ub was mixed with HOIL pre-incubated with increasing

concentrations of M1 or K63 di-Ub, for final concentrations of 1 μM HOIL-1, 10 μM E2-Ub and 0–40 μM di-Ub. To perform RNF216 E2-Ub discharge for allosteric di-Ub EC50 calculations, E2-Ub was mixed with RNF216 pre-incubated with increasing concentrations of K63 di-Ub, for final concentrations of 4 μM RNF216, 40 μM E2-Ub and 0–40 μM di-Ub. Reactions were incubated for 1 min at room temperature and quenched by addition of equal volume of 2x LDS buffer without reducing agent.

Samples were resolved by SDS-PAGE using 4–12% Bis-Tris gels (NuPAGE, Invitrogen). Gels were stained with Coomassie brilliant blue (InstantBlue, Abcam) and imaged on a ChemiDoc imaging systems (Bio-Rad).

### Maltose ubiquitination assay and MALDI-TOF analysis of reaction products

Reactions containing 200 nM UBE1, 2 μM UbcH7, 10 μM Ub and 20 mM maltose were set up in DPBS buffer pH 7.5 supplemented with 10 mM MgCl2 and 0.5 mM TCEP. Indicated E3 was added into the reaction to a final concentration of 5 μM. Time 0 samples were taken by mixing with 2x LDS buffer supplemented with 0.2 M DTT. Reactions were initiated by addition of 10 mM ATP and incubated for the specified time. Reactions were stopped by adding 2x LDS buffer. Samples were resolved by SDS-PAGE using 12% Bis-Tris gel (NuPAGE, Thermo Fisher). Gels were stained with Coomassie brilliant blue (InstantBlue Abcam) and imaged with ChemiDoc imaging systems (Bio-Rad). In Fig. 7F, blocked M1 di-Ub (GPG-di-UbΔC-term Gly) was used to allosterically activate HOIL-1 activity. Reactions were incubated at room temperature.

For MALDI-TOF, the method described by Signor et al. was used to deposit reaction samples onto MALDI target plate (MSP 96 target, Bruker)[73]. In brief, matrices containing 20 mg/mL 2,5-dihydroxybenzoic acid (DHB) and 20 mg/mL α-Cyano-4-hydroxycinnamic acid (CHCA) were made separately and mixed in 1:1 ratio to generate the DHB-CHCA matrix. Samples diluted in acetonitrile and 5% formic acid in 7:3 (v:v) ratio were mixed in 1:1 ratio with the DHB-CHCA matrix and deposited onto the target plate. Spectra were acquired manually on a Bruker microflex using 30% laser power (60 Hz) at a detection gain of 4.4 using flexAnalysis 3.4 software (Bruker).

### HOIP chain formation assay

M1-linked Ub chain formation assays used to probe the overall HOIP E3 ligase activity were performed as previously described[6]. 200 nM E1, 1 μM E2 (UbcH5B), 1 μM HOIP RBR and 40 μM Ub with native N- and C-termini were incubated in 50 mM HEPES pH 7.9, 100 mM NaCl, 10 mM MgCl2, 0.6 mM DTT, and 10 mM ATP in the presence or absence of 10 μM fully blocked M1-linked di-Ub or fully blocked I44A M1-linked di-Ub at 30 °C. Reactions were stopped at different time points by adding samples to 2x SDS-sample buffer with DTT and heating for 5 min to 95 °C. Samples were analysed on 4–12% Bolt Bis-Tris SDS-PAGE gels (Invitrogen) and visualized using Coomassie Brilliant blue dye.

### HOIP-Ub thioester formation assay

The HOIP-Ub thioester formation assay used to probe the first step of the HOIP RBR reaction is based on previously described methods[6]. 100 nM E1, 2 μM E2 (UbcH5B), and 8 μM Ub were incubated in 50 mM HEPES pH 7.9, 100 mM NaCl, 10 mM MgCl2, and 5 mM ATP for 5 min at 25 °C to pre-load E2 with Ub. HOIP RBR was preincubated with blocked di-Ub (GPG-di-UbΔC-term Gly) for 5 min at 25 °C and added to the loaded E2-Ub mixture to final concentrations of 1 μM HOIP RBR. Reactions were stopped after 30 s by mixing samples 1:1 v/v with pre-heated 2x SDS loading buffer without DTT, analysed on 4–12% Bolt Bis-Tris SDS-PAGE gels (Invitrogen) and visualized using Coomassie Brilliant blue dye. Gels were scanned on a Li-COR Odyssey scanner using the 700 nm channel and bands quantified using ImageStudio software (LI-COR). The amount of HOIP-Ub thioester was determined as the

fraction of HOIP-Ub to total HOIP for each sample. Data were plotted and analysed in GraphPad Prism.

## HOIP-Ub transfer assay

To measure the second step of the HOIP RBR reaction, we developed a HOIP-Ub transfer assay. For this, we first formed an E2-Ub conjugate by incubating 100 nM E1, 4 μM UbcH5B, and 8 μM N-terminally blocked Ub (GPG-Ub) in 50 mM HEPES pH 7.9, 100 mM NaCl, 10 mM MgCl$_2$, and 5 mM ATP for 10 min at 25 °C. Loading of the E2 was stopped by adding apyrase (Sigma-Aldrich A6410, 25 mU/μl final concentration) and incubation for 5 min at 25 °C. Ub transfer from the E2 to the HOIP RBR was induced by addition of 8 μM HOIP RBR and incubation for 100 s at 25 °C, followed by 20 s on ice. Ub transfer from HOIP RBR to the substrate (TAMRA-Ub) was performed on ice and induced by addition of 4 μM fluorescent TAMRA-Ub (LifeSensors SI270T, Ub labelled with a single TAMRA molecule at a defined location avoiding the N-terminus and the lysine sidechains) in the presence or absence of 20 μM fully blocked M1-linked di-Ub. Reactions were stopped at indicated timepoints by mixing samples with 2x SDS loading buffer without DTT. Samples were analysed on 12% Bolt Bis-Tris SDS–PAGE gels (Invitrogen). Gels were first scanned for the TAMRA-Ub signal on a FLA-5100 fluorescent image analyser (Fujifilm) using a 532 nm laser and a 575 nm filter, followed by staining with Coomassie Brilliant blue dye and scanning on a Li-COR Odyssey scanner using the 700 nm channel. Fluorescent band intensities were quantified using the Multi Gauge software (Fujifilm) and Coomassie bands were quantified using ImageStudio software (LI-COR). The remaining HOIP-Ub thioester was calculated as the fraction of HOIP-Ub to total HOIP for each sample.

## Statistics

Intensities of Coomassie stained SDS-PAGE gel bands were quantified in Image Lab 6.1 (Bio-Rad) or Image Studio 5.2.5 (Li-COR Biosciences). Data were analysed and plotted using GraphPad Prism 9. Biochemistry experiments were performed at least in triplicate unless otherwise stated, and representative gels are shown. Details of statistical tests and replicates are given in the figure legends.

## Reporting summary

Further information on research design is available in the Nature Portfolio Reporting Summary linked to this article.

## Data availability

The study made use of the following publicly available data sets: PDB entries: 1UBQ, 2HAP, 4AP4, 4BM9, 4Q5E, 5EDV, 6DJW, 7B5L, 7M4M, 7M4O and 7V8F; AlphaFold DB models Q9BYM8 (HOIL-1) and F8WDI8 (RNF216). Atomic structures and diffraction data generated in this study have been deposited in the PDB under accession codes 8EAZ (HOIL-1(C460A)/UbcH7(C86K)-Ub/Ub) and 8EB0 (RNF216(C688A)/UbcH7(C86K)-Ub/Ub). All other data supporting the conclusions are available in the article. Biochemical data generated in this study are provided in the Source data file. Source data are provided with this paper.

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

## Acknowledgements

We thank Simon Cobbold (WEHI) for help with MALDI data collection and analysis, David Komander (WEHI) for helpful comments and suggestions, Janet Newman (Commonwealth Scientific and Industrial Research Organisation [CSIRO] Collaborative Crystallisation Centre [C3]) and Geoffrey Kong (Monash Macromolecular Crystallisation Facility) for help with crystallisation, and Dale Calleja (WEHI) for help with RNF216

diffraction data collection. This work was funded by WEHI, National Health and Medical Research Council (NHMRC) Ideas Grant 1182757 (to B.C.L.), NHMRC Investigator Grant 1195038 (to J.S.) and through Victorian State Government Operational Infrastructure Support and Australian Government NHMRC IRIISS grant 9000719 (to J.S.). X.S.W. is supported by an Australian Government Research Training Program Scholarship. We acknowledge the support of the Bodhi Education Foundation. The mass spectrometry analysis was performed at the WEHI Proteomics Laboratory. This research was undertaken in part using the MX2 beamline at the Australian Synchrotron, part of the Australian Nuclear Science and Technology Organisation (ANSTO), and made use of the Australian Cancer Research Foundation (ACRF) detector.

## Author contributions

X.S.W. and T.R.C. designed and performed experiments and interpreted data. S.J.T. and L.W.R. helped with protein expression and purification and performed experiments. W.T.L. performed experiments under the supervision of X.S.W. X.S.W., T.R.C. and B.C.L. wrote the manuscript. J.S. co-supervised, X.S.W. helped interpret results, and commented on the manuscript. B.C.L. designed the overall study, supervised the work, interpreted the results, and secured funding.

## Competing interests

The authors declare no competing interests.
