## [Peer Review File · Nature Communications]

The unifying catalytic mechanism of the RING-between-RING E3 ubiquitin ligase familyReviewer #1 (Remarks to the Author):

Much of the understanding of RBR E3 catalytic mechanism has derived from the structural and biochemical studies of HOIP, PARKIN and HHARI. In this manuscript, Wang et al. characterized the structures and biochemical reactions of two RBR E3s, HOIL-1 and RNF216, to further define the RBR catalytic mechanism. The study showed that HOIL-1 and RNF216 are activated allosterically by di-Ub (M1 and K63 for HOIL-1 and K63 for RNF216). These findings are in agreement with published works from Lechtenberg's group and others. The allosteric activation by di-Ub accelerated the transthiolation reaction but not the aminolysis step. Wang et al. initially attempted to crystallize the di-Ub bound RBR-E2-Ub complexes to gain insight into how di-Ub allosterically activates the transthiolation reaction. However, they failed to obtain crystals with di-Ub and only obtained the structures with mono-Ub bound. Both structures of HOIL-1 and RNF216 RBR domain bound to E2-Ub and a mono-Ub resemble previously published structures of other RBR/E2-Ub/Ub or RBR/E2/Ub transthiolation complexes from HOIP, HHARI and PARKIN. The binding interfaces were validated via mutagenesis analyses and biochemical assays. The RING1-Ub and RING2/E2-Ub interactions are mostly conserved across different RBR E3s. The core allosteric Ub binding site is conserved where Ub binding induces straightening of RING1-IBR helix as seen previously. Although the structures lack allosteric di-Ub, biochemical analysis suggested that both proximal and distal Ub subunits of M1-di-Ub bind HOIL-1. Proximal Ub binds the allosteric Ub site and the distal Ub likely binds a helix at the N-terminus of RBR domain. In contrast, RNF216 prefers the distal Ub of K63-di-Ub and its N-terminal helix of RBR domain does not contribute to binding. A novel domain fold of Zn₂/Cys6 binuclear cluster was identified in HOIL-1 RING2 domain. The role of this cluster remains unclear. It was suggested that this feature may serve to accommodate "substrate" of HOIP. A recent published non-proteinaceous substrate maltose was tested. Surprisingly RNF216 and HOIP were able to ubiquitinate maltose suggesting that Zn₂/Cys6 of HOIL-1 is not a specific polysaccharide substrate binding platform. A histidine H510 at the C-terminus of Zn₂/Cys6 cluster was shown to be important for HOIL-1-mediated Ub transfer to substrate.

Overall, this is a technically sound study. The manuscript is well-written and clear. Unfortunately the structures of the mono-Ub bound transthiolation complexes did not unveil novel mechanistic insight compared to the prior RBR E3/E2-Ub/Ub complexes. Nevertheless the study moves the field forward by showing that the catalytic RBR mechanism is conserved with some variations in the allosteric activation by di-Ub.

Comments:

Lechtenberg's group recently shown that mono-Ub did not activate RNF216 transthiolation reaction (Cotton et al. Mol Cell 2022). However, the structure obtained here contained mono-Ub and the RING1-IBR helix adopts an activated straight conformation suggesting that the complex likely resembles the "activated" state. Please explain this discrepancy. This reviewer noticed that the concentration of protein complex used for protein crystallization contained high micromolar concentration of mono-Ub. Could the high mono-Ub concentration stimulate the transthiolation reaction?

The role of Zn₂/Cys6 cluster is preliminary and not well-developed in this study. Suggestion of its role in atypical substrate specificity in the abstract is misleading. There is scope to investigate this further and could be interesting, but this is probably beyond the theme of the current manuscript.

The refinement statistic, R_{free}, for RNF216 complex could be improved further.

Minor comment

Page 9 line 5 RING-IBR should read RING1-IBR

Reviewer #2 (Remarks to the Author):

The manuscript by Wang et al. described structural and functional characterizations of HOIL-1 and RNF216, both of which are RBR E3 ligases. They used X-ray crystallography to determine the quaternary structure of HOIL-1L/UbcH7~Ub/M1 di-Ub and that of RNF216/UbcH7~Ub/mono-Ub. These crystal structures revealed a conserved RBR RING1 interaction with E2 associated with the first transthiolation step. Biochemical assays showed that activation of HOIL-1 is stimulated by M1 di-Ub and to a lesser extent by K63 di-Ub; the same assays showed that RNF216 is activated by K63 di-Ub and not by M1 di-Ub or other linkage types. The structural findings were supported by a series of functional assays with site-directed mutagenesis based to verify the functional contributions of individual intermolecular interactions. These systematic analyses elegantly illustrated how RBR E3 ligases use the same structural motifs to interact with E2 and how their functionalities are allosterically activated by ubiquitin moieties connected by different linkage types. Perhaps the most exciting finding in this study is the identification of an unusual Zn²/Cys6 binuclear cluster to coordinate two zinc ions in the RING2 domain of HOIL-1. The structure also revealed a potential catalytic site formed by C460 and H510 to achieve the non-proteinaceous ubiquitination of polysaccharides, including maltose. The catalytic activity of maltose ubiquitination and the functional contributions of C460 and H510 were further verified by mutagenesis. Collectively, this manuscript provides comprehensive characterizations of the structure-function relationships of HOIL-1 and RNF216 to broaden our understanding of the RBR E3 family. I would support the publication of this manuscript in Nature Communications. Nevertheless, there are quite a few technical issues and specific descriptions that should be addressed before it is accepted for publication.

Major comments:

1. Can the authors provide the structural basis of substrate recognition of the unusual Zn²/Cys6 binuclear cluster in the context of carbohydrate ubiquitination?
2. Page 5, line 1: It appears that K48 di-Ub also enhances HOIL-1 binding to UbcH7(C86K)-Ub, although to a less extent than does M1 di-Ub. This enhancement does not translate into the functional activation of HOIL-1's E2-Ub discharging activity. How could the discrepancy be explained?
3. Page 5, line 30: The authors stated that SEC-MALS indicated stoichiometric complex formation of UbcH7-Ub with HOIL-1 or RNF216 with or without the allosteric activator, M1 di-Ub or K63 di-Ub. However, sizable discrepancies are observed for these complexes that amount to approximately 50% less of the molecular mass of UbcH7-Ub, suggesting that UbcH7-Ub binding is weak, and the quaternary complexes are not very stable. The relatively weak binding is manifested in asymmetric SEC elution peaks of the corresponding complexes. The authors are therefore advised to revise the statement on Page 6, line 1, that the formation of stable 1:1:1 complexes were observed.
4. The high B-factors of UbcH7 in the reported crystal structures (Supplementary Fig. 3) result in fragmented electron density for UbcH7. In the Materials and Methods section (Page 18-19), the authors refer to the use of a high-resolution UbcH7 as a reference model for refinement. Given the relatively low resolutions of the crystal structures and the high flexibility of UbcH7 within the crystals, can the authors provide additional information to indicate the reliability of the atomic models used to describe the intermolecular interactions with UbcH7? Indeed, many of the observed interactions have been verified biochemically aided by mutagenesis, but it would still be useful to include additional descriptions on how the protein interfaces pertinent to UbcH7 are modeled.
5. Page 8, last paragraph: When describing the interactions associated with the RING2 domains of HOIL-1 and RNF216, the authors mentioned fewer interactions and most of the binding surface being attributed to the donor Ub. Could the authors quantify these by changes in solvent accessible surface area (SASA) and estimate their energetic contributions by using ePISA?
6. Page 9, line 10: ITC analysis indicates that WT HOIL-1 binds to M1 di-Ub with a K_d of 50 μM, which is very weak for protein-protein interactions. What is the biological relevance of such weak interaction in vivo?
7. Page 9, line 18: The authors speculate that D384 and D385 of HOIL-1 may contribute

to the binding to the allosteric ubiquitin. It should be possible to generate a triple mutant - E383A/D384A/D385A - to verify such a statement.

8. Figure 2: The tabulated ITC result of HOIL-1/M1 di-Ub has two entries with different KD values, 0.5 and 0.2 μ M. What is the second one that is not described in the manuscript? Additionally, several molar ratios (n values) clearly deviated from unity, 1.22, 0.71 and 0.78. Could this be inaccurate protein concentration estimation or significant proportions of inactive binding partners?

9. The SDS-PAGE of Figure 1d exhibits significant amount of high-molecular-weight material in HOIL-1 helix-RBR that resembles NZF-RBR. In the reconstitute biochemical assays, the individual protein components are expected to be highly purified. What can be the high-molecular-weight material in this SDS-PAGE?

10. Page 6, last paragraph and Supplementary Fig. 3e: What could be the contributing factors of spontaneous degradation of M1 di-Ub into mono-ubiquitin in the crystals of HOIL-1/UbcH7~Ub/M1 di-Ub?

Minor comments:

1. Page 4, line 32: An error estimate of the dissociation constant for HOIL-1 binding to UbcH7(C86K)-Ub, stimulated by M1 di-Ub, was provided, $K_D = 0.35 \pm 0.15 \mu\text{M}$, but $n=2$. If the error estimate is RMSD, the minimum sample size should be 3.

2. Page 4, line 29: A mutant of UbcH7(C86K) is used in the binding assay instead of wild-type UbcH7, but the corresponding labels in Fig. 2 do not specify the C86K mutation (but the corresponding caption also uses the same description, UbcH7(C86K)). The nomenclature should be used consistently or defined in the manuscript when abbreviated/simplified forms are used.

3. di-Ub is labeled as Di-Ub in Figure 1, Supplementary Figure 2c (Table) and other places.

4. Page 2, line 32: NEDD should be capitalized.

Reviewer #4 (Remarks to the Author):

The manuscript by Wang et al. describes a structural and biochemical workup of two Ring-between-Ring (RBR) ubiquitin ligases, HOIL-1 and RNF216. RBRs are unique ubiquitin ligases in that they catalyze ubiquitylation by first binding to an E2~ubiquitin molecule and promoting the transfer of ubiquitin from the ubiquitin-conjugating enzyme E2 to the RING 2 domain of the ligase, and then ultimately to a protein substrate bound to the ligase. Several RBRs including PARKIN, ARIH1(HHARI) and HOIP have had their molecular mechanisms described in detail, whereas those of HOIL-1 and RNF216 still awaited discovery and was the main thrust of this paper. X-ray crystal structures of fragments for both ligases are presented and buttressed by a suite of biochemical experiments ranging from isothermal calorimetry, size exclusion chromatography coupled with multi-angle light scattering, and numerous kinetic ubiquitylation assays. The results show that important aspects of HOIL-1 and RNF216 function, including E2~ubiquitin binding to the ligase, transfer of ubiquitin from E2 to the RING2 domain, and the binding of an allosteric ubiquitin or ubiquitin-like activator, are similar as existing models for RBRs in the literature. Specifically, the addition of free di-ubiquitin results in the stimulation of E2~ubiquitin binding to both ligases and in a linkage-specific manner. Mutants of both structures were biochemically characterized to confirm various aspects of the model. While HOIL-1 and RNF216 appear to function similarly as the other RBRs mentioned here, differences were also observed in the details and are described in the manuscript. A model is presented where protein substrate bound to RBR becomes modified with a poly-ubiquitin chain which then allosterically enhances ligase activity through a feed-forward mechanism.

This well-written paper was a pleasure to read and is packed with useful information regarding RBR ubiquitin ligase function. While the results lead to models that are clearly similar to those in the literature for better characterized E3s, this should not take away from the impact of the work, as our understanding of RBRs in general is still lacking. As

such, this study is a major advance and should be published in Nature Communications, though we suggest the following changes prior to that.

Major points:

While the feed forward mechanism is interesting, there are a couple of aspects that make the model challenging to accept, at least for now. First, the protein substrate must receive at minimum 2 ubiquitins to generate the di-ubiquitin chain, although since the substrate binding site is not mentioned or shown, it very well may be the case that far more ubiquitins are necessary for the chain to reach the allosteric site. As such, it is unclear what the functional advantage would be in a case where several ubiquitins are first transferred to protein substrate prior to engagement of the allosteric mechanism for enhanced activity. Another issue with the model is that the authors show here that E2~ub binding is weak in the absence of the allosteric activator, perhaps even too weak for E2~ub to associate with the RBR-substrate complex in cells where UBC7 levels are relatively low (indeed the authors were not even able to measure an equilibrium dissociation constant for E2 and HOIL-1 in the absence of di-ubiquitin). As such, perhaps this detailed model should be removed from the manuscript unless far more substantial evidence can be produced.

Minor points:

Supplementary Figure 1b: the experiment is designed to follow the effect of the presence of di-ubiquitin on the stimulation of HOIL-1 thioester bond formation with ubiquitin. However, we wonder how the authors distinguish this from isopeptide bond formation between HOIL-1 and ubiquitin. Comparison of reducing and non-reducing gels would be helpful here.

Figure 7e: it is speculated that the mutation of His 510 to Ala in HOIL-1 stabilizes the thioester bond between the mutant ligase and ubiquitin as evidenced by the appearance of a slowly migrated band in the gel. While this is compelling, the presence of doublet bands makes interpretation of the result more challenging (in other words, if this really is HOIL-1~cys-ubiquitin, why are there two bands that disappear upon treatment with reducing agent). Also, it would be helpful to have both a non-reducing and reducing gel for the WT ligase for the sake of comparison.

A defect is reported for H510A HOIL-1 in maltose ubiquitylation. However, this is potentially a non-physiological substrate and as such, its relevance to HOIL-1 function is still poorly understood. We suggest addressing this by adding an experiment comparing the activity of H510A to wild-type in the aminolysis assay with TAMRA labeled ubiquitin.

REVIEWER COMMENTS

We thank the reviewers for their general positive feedback and valuable suggestions. We address the reviewers' comments point-by-point below. We reproduce the original reviewers' comments in black font with our replies in blue font.

Reviewer #1 (Remarks to the Author):

Much of the understanding of RBR E3 catalytic mechanism has derived from the structural and biochemical studies of HOIP, PARKIN and HHARI. In this manuscript, Wang et al. characterized the structures and biochemical reactions of two RBR E3s, HOIL-1 and RNF216, to further define the RBR catalytic mechanism. The study showed that HOIL-1 and RNF216 are activated allosterically by di-Ub (M1 and K63 for HOIL-1 and K63 for RNF216). These findings are in agreement with published works from Lechtenberg's group and others. The allosteric activation by di-Ub accelerated the transthiolation reaction but not the aminolysis step. Wang et al. initially attempted to crystallize the di-Ub bound RBR-E2-Ub complexes to gain insight into how di-Ub allosterically activates the transthiolation reaction. However, they failed to obtain crystals with di-Ub and only obtained the structures with mono-Ub bound. Both structures of HOIL-1 and RNF216 RBR domain bound to E2-Ub and a mono-Ub resemble previously published structures of other RBR/E2-Ub/Ub or RBR/E2/Ub transthiolation complexes from HOIP, HHARI and PARKIN. The binding interfaces were validated via mutagenesis analyses and biochemical assays. The RING1-Ub and RING2/E2-Ub interactions are mostly conserved across different RBR E3s. The core allosteric Ub binding site is conserved where Ub binding induces straightening of RING1-IBR helix as seen previously. Although the structures lack allosteric di-Ub, biochemical analysis suggested that both proximal and distal Ub subunits of M1-di-Ub bind HOIL-1. Proximal Ub binds the allosteric Ub site and the distal Ub likely binds a helix at the N-terminus of RBR domain. In contrast, RNF216 prefers the distal Ub of K63-di-Ub and its N-terminal helix of RBR domain does not contribute to binding. An novel domain fold of Zn₂/Cys₆ binuclear cluster was identified in HOIL-1 RING2 domain. The role of this cluster remains unclear. It was suggested that this feature may serve to accommodate "substrate" of HOIP. A recent published non-proteinaceous substrate maltose was tested. Surprisingly RNF216 and HOIP were able to ubiquitinate maltose suggesting that Zn₂/Cys₆ of HOIL-1 is not a specific polysaccharide substrate binding platform. A histidine H510 at the C-terminus of Zn₂/Cys₆ cluster was shown to be important for HOIL-1-mediated Ub transfer to substrate.

Overall, this is a technically sound study. The manuscript is well-written and clear. Unfortunately the structures of the mono-Ub bound transthiolation complexes did not unveil novel mechanistic insight compared to the prior RBR E3/E2-Ub/Ub complexes. Nevertheless the study moves the field forward by showing that the catalytic RBR mechanism is conserved with some variations in the allosteric activation by di-Ub.

Comments:

Lechtenberg's group recently shown that mono-Ub did not activate RNF216 transthiolation reaction (Cotton et al. Mol Cell 2022). However, the structure obtained here contained mono-Ub and the RING1-IBR helix adopts an activated straight conformation suggesting that the complex likely resembles the "activated" state. Please explain this discrepancy. This reviewer noticed that the concentration of protein complex used for protein crystallization contained high micromolar concentration of mono-Ub. Could the high mono-Ub concentration stimulate the transthiolation reaction?

RESPONSE: We agree with the reviewer that our crystal structures show the activated state of the RBR ligases, even though only mono-Ub is visible/present in the structures. We previously reported similar observations with the RBR ligase HOIP (Lechtenberg *et al.*, Nature 2016).

In our view, the Ub moiety that binds to the allosteric site in the RING1-IBR helix/IBR interface and is visible in our structures is responsible for inducing the conformational change in the helix, whereas the 2nd ubiquitin moiety of the di-Ub contributes avidity and chain-type specificity.

We now add a new experiment in Supplementary Fig. 5d that shows that mono-Ub can activate HOIL-1 at a very high (200 μ M) concentration by \sim 2-fold compared to HOIL-1 in the absence of an activator, but to only \sim 20% of the activity of HOIL-1 in the presence of 40-time less (5 μ M) M1-linked di-Ub. This new experiment together with the other data in Fig. 6 and Supplementary Fig. 5 of our manuscript (allosteric site mutant HOIL-1 I358R, I44A di-Ub mutants) further refines our model and in our view resolves the apparent discrepancy raised by the reviewer.

We also tried to perform the same experiment with RNF216, but this was confounded by multiple issues. First, the difference of RNF26 activity in the presence or absence of K63 di-Ub is much smaller than for HOIL-1, making it difficult to monitor small changes in activity. We observe a weak activating effect of 200 μ M Ub, but this is confounded by a significant formation of di-Ub during the experiment, which will robustly activate RNF216 independently even compared to the higher concentration of the mono-Ub. We see reduced di-Ub formation when we use K11R/K63R mutant mono-Ub as the activator (K63 and K11 are the main chain-types formed by RNF216), but also a reduced activating effect. At this point we cannot identify if the difference between WT and K11R/K63R Ub is due to the differences in di-Ub formation or other effects by the K11R/K63R mutations. Due to these complications, we decided to not include this experiment in the manuscript at this point.

The role of Zn²⁺/Cys6 cluster is preliminary and not well-developed in this study. Suggestion of its role in atypical substrate specificity in the abstract is misleading. There is scope to investigate this further and could be interesting, but this is probably beyond the theme of the current manuscript.

RESPONSE: We fully agree with the reviewer that further investigation of the function of the Zn²⁺/Cys6 cluster would be very interesting. This is ongoing work in the lab. However, we agree with the reviewer that this is outside the scope of the current manuscript that focuses more on the general RBR mechanism.

We revised the wording in the abstract regarding the role of the Zn²⁺/Cys6 cluster. The last sentence of the abstract now reads: "We finally identify that the HOIL-1 RING2 domain contains an unusual Zn²⁺/Cys6 bivalent zinc cluster that is required for catalytic activity and may represent a specific adaptation for substrate specificity." Our data show that, in particular, His510 of the Zn²⁺/Cys6 bivalent cluster is required for HOIL-1's 2nd catalytic step. We also think it is justified to speculate that the Zn²⁺/Cys6 cluster may play an important role in HOIL-1 substrate specificity (may it be polysaccharides or Ser/Thr sidechains in proteins), given its proximity to the active site, and the fact that additional ZF insertions in the RBRs HOIP and RNF216 have been shown to mediate substrate specificity as outlined in our discussion.

The refinement statistic, R_{free}, for RNF216 complex could be improved further.

RESPONSE: We agree with the reviewer that the refinement statistics for the RNF216 complex are not what one would normally expect from a structure at 3.03 Å resolution. We spent extensive time and effort on the model building and refinement for our initial submission and structure deposition,

trying many different strategies without yielding further improvement of the statistics. We now revisited the refinement strategy again without any further improvement. One major factor for the poor statistics likely is the poor density for the E2, which contributes more than 25% of all amino acids in our structure.

We clearly discuss and present these limitations in the main text, Methods, Table 1, and Supplementary Fig. 3. They are also evident from the PDB validation report and the “sliders graphic” that will be posted alongside our structure once the PDB website releases our structures.

Please also note our reply to Reviewer #2, major point 4, for additional discussions on the quality of the crystal structures.

Minor comment

Page 9 line 5 RING-IBR should read RING1-IBR

Thanks, we corrected this typo, as well as a few other instances.

Reviewer #2 (Remarks to the Author):

The manuscript by Wang et al. described structural and functional characterizations of HOIL-1 and RNF216, both of which are RBR E3 ligases. They used X-ray crystallography to determine the quaternary structure of HOIL-1L/UbcH7~Ub/M1 di-Ub and that of RNF216/UbcH7~Ub/mono-Ub. These crystal structures revealed a conserved RBR RING1 interaction with E2 associated with the first transthiolation step. Biochemical assays showed that activation of HOIL-1 is stimulated by M1 di-Ub and to a lesser extent by K63 di-Ub; the same assays showed that RNF216 is activated by K63 di-Ub and not by M1 di-Ub or other linkage types. The structural findings were supported by a series of functional assays with site-directed mutagenesis based to verify the functional contributions of individual intermolecular interactions. These systematic analyses elegantly illustrated how RBR E3 ligases use the same structural motifs to interact with E2 and how their functionalities are allosterically activated by ubiquitin moieties connected by different linkage types. Perhaps the most exciting finding in this study is the identification of an unusual Zn2/Cys6 binuclear cluster to coordinate two zinc ions in the RING2 domain of HOIL-1. The structure also revealed a potential catalytic site formed by C460 and H510 to achieve the non-proteinaceous ubiquitination of polysaccharides, including maltose. The catalytic activity of maltose ubiquitination and the functional contributions of C460 and H510 were further verified by mutagenesis. Collectively, this manuscript provides comprehensive characterizations of the structure-function relationships of HOIL-1 and RNF216 to broaden our understanding of the RBR E3 family. I would support the publication of this manuscript in Nature Communications. Nevertheless, there are quite a few technical issues and specific descriptions that should be addressed before it is accepted for publication.

Major comments:

1. Can the authors provide the structural basis of substrate recognition of the unusual Zn2/Cys6 binuclear cluster in the context of carbohydrate ubiquitination?

RESPONSE: We agree with the reviewer that the unique Zn2/Cys6 binuclear cluster in HOIL-1's RING2 is an exciting finding that warrants further investigation. As we state in the text, we initially hypothesised that this feature may be critical for HOIL-1's ability to ubiquitinate polysaccharides. However, our *in vitro* experiments with RNF216, which does not feature a Zn2/Cys6 binuclear cluster but can still ubiquitinate maltose under the same conditions as HOIL-1, suggest that the Zn2/Cys6 cluster is not strictly required for polysaccharide ubiquitination *in vitro*. In our view, careful cellular

experiments under more physiological conditions are required to investigate if and under which conditions HOIL-1 (and potentially other RBR E3 ligases) ubiquitinates polysaccharides or other potential substrates in biological settings. This is ongoing work that requires the establishment of many new research tools and in our view is outside the scope of the current manuscript. Even though we cannot fully resolve the mechanism and observed activities in this study, we believe these results are important for the field as they put the recent finding by Kellsall *et al.* (*EMBO J* 2022, ref. 24) into perspective and will provide the structural basis for subsequent work in this exciting emerging field of non-lysine ubiquitination.

2. Page 5, line 1: It appears that K48 di-Ub also enhances HOIL-1 binding to UbcH7(C86K)-Ub, although to a less extent than does M1 di-Ub. This enhancement does not translate into the functional activation of HOIL-1's E2-Ub discharging activity. How could the discrepancy be explained?

RESPONSE: The ITC experiments and E2-Ub discharge experiments are fundamentally different experiments with different readouts and are performed under different experimental conditions, in particular vastly different protein concentrations. We use 1 μM HOIL-1 and 5 μM di-Ub in the E2-Ub discharge assays. The ITC experiments use 50 μM HOIL-1 and 150 μM di-Ub. The EC_{50} of M1 di-Ub for the activation of HOIL-1 is 8 μM (see Fig. 1c), therefore the concentration used in the ITC experiments is almost 20 times above the EC_{50} .

The fundamental differences in the two experiments are also reflected in the apparent discrepancy between the activating effects of M1 di-Ub in the E2-Ub discharge assay where we determine an EC_{50} of 8 μM (Fig. 1c) compared to the K_D of 50 μM as determined by ITC (Supplementary Fig. 5a).

Based on the binding modes observed in the crystal structures, we propose that even mono-Ub may have weak affinity for the allosteric binding site, and therefore activate Ub at very high concentrations. In line with this, and also in response to comments made by reviewer 1, major point 1, we now include new data that shows that mono Ub can activate HOIL-1 at high concentrations. In the presence of 200 μM mono-Ub, HOIL-1 is ~2-times more active than in the absence of allosteric activator, but only ~ 20% as active as in the presence of 40 times less (5 μM) M1-linked di-Ub (new Supplementary Figure 5d). We hypothesise that the allosteric Ub molecule observed in our structures is most critical for the activating effect, whereas binding of the 2nd Ub molecule adds avidity and specificity. This explains why K48 di-Ub may have a weak effect at high concentrations in the ITC. One of the two Ub protomers in the K48 di-Ub may bind to the allosteric site with weak affinity but lacking the enhanced avidity of the 2nd Ub molecule of an M1-linked di-Ub.

3. Page 5, line 30: The authors stated that SEC-MALS indicated stoichiometric complex formation of UbcH7-Ub with HOIL-1 or RNF216 with or without the allosteric activator, M1 di-Ub or K63 di-Ub. However, sizable discrepancies are observed for these complexes that amount to approximately 50% less of the molecular mass of UbcH7-Ub, suggesting that UbcH7-Ub binding is weak, and the quaternary complexes are not very stable. The relatively weak binding is manifested in asymmetric SEC elution peaks of the corresponding complexes. The authors are therefore advised to revise the statement on Page 6, line 1, that the formation of stable 1:1:1 complexes were observed.

RESPONSE: We thank the reviewer for pointing out this inaccuracy. We revised the statement to reflect this. The statement (now on p. 6, l. 2) now reads: "The SEC-MALS derived molecular mass calculations support formation of 1:1:1 ternary complexes of the HOIL-1 or RNF216 RBR modules with an UbcH7(C86K)-Ub conjugate and allosteric di-Ub, although a ~20–25% discrepancy between the calculated and observed molecular weight of the ternary complexes suggests UbcH7(C86K)-Ub binding is relatively weak under these conditions."

4. The high B-factors of UbCH7 in the reported crystal structures (Supplementary Fig. 3) result in fragmented electron density for UbCH7. In the Materials and Methods section (Page 18-19), the authors refer to the use of a high-resolution UbCH7 as a reference model for refinement. Given the relatively low resolutions of the crystal structures and the high flexibility of UbCH7 within the crystals, can the authors provide additional information to indicate the reliability of the atomic models used to describe the intermolecular interactions with UbCH7? Indeed, many of the observed interactions have been verified biochemically aided by mutagenesis, but it would still be useful to include additional descriptions on how the protein interfaces pertinent to UbCH7 are modeled.

RESPONSE: As the reviewer points out, we aimed to be very upfront with discussing the limitations of our crystal structures, allowing the reader to clearly understand what detail can be derived from our structures. As the reviewer acknowledges, we additionally verified many of the key interactions biochemically.

To further clarify these details, we now specified in the Methods how the refinement program phenix.refine handles reference models: "We therefore used a high-resolution UbCH7 structure (PDB 7V8F ref. 68) to generate dihedral angle reference model restraints during refinement in phenix.refine⁶⁹." (p. 20, l. 4), and include an additional reference (Headd *et al.*, Acta Cryst D, 2012, ref. 69) that describes these methods in phenix. We also more clearly state in the main text that the sidechain densities for the UbCH7 molecule are weak: "The latter features gaps in electron density and relatively weak density and high B-factors especially for the UbCH7 molecule (Supplementary Fig. 3b,c), allowing us to confidently place individual proteins and domains but providing limited information at the amino acid level, in particular the amino acid sidechains." (p. 6, l. 22; new addition underlined).

5. Page 8, last paragraph: When describing the interactions associated with the RING2 domains of HOIL-1 and RNF216, the authors mentioned fewer interactions and most of the binding surface being attributed to the donor Ub. Could the authors quantify these by changes in solvent accessible surface area (SASA) and estimate their energetic contributions by using ePISA?

RESPONSE: We now report solvent accessible surface area analysis calculated using ChimeraX. These analyses show that for both complexes, donor Ub contributes a greater interaction surface with the helix-RING2 module (HOIL-1=968 Å², RNF216 = 1211 Å²), than UbCH7 (HOIL-1=374 Å², RNF216 = 522 Å²). We now include these data in Figure 5a,b, added a reference to these figure panels in the last paragraph on page 8, and include details on the calculations in the Methods section. Due to the relatively low resolution of the crystal structures and therefore poor certainty in the amino acid sidechain positions (see previous comment), we refrain from performing any more detailed analysis, e.g., energetic contributions via ePISA.

6. Page 9, line 10: ITC analysis indicates that WT HOIL-1 binds to M1 di-Ub with a K_D of 50 μM, which is very weak for protein-protein interactions. What is the biological relevance of such weak interaction in vivo?

RESPONSE: It is correct that a K_D of 50 μM is weak for general protein-protein interactions. However, an affinity of 50 μM is not out of the ordinary for Ub interactions that are often weak, in the low-to-mid μM range (see e.g., Hurley *et al.*, Biochem J, 2006, <https://www.ncbi.nlm.nih.gov/pmc/articles/PMC1615911/>). Ub interactions are often enhanced by avidity effects due to longer Ub chains or Ub chains attached to a substrate that may be part of the same complex as the binding partner. In line with the first argument, Kellsall *et al.* (EMBO J, 2022, ref. 24) recently reported that M1-linked tetra-Ub has a stronger activating effect compared to M1-linked

di-Ub, suggesting that M1-linked tetra-Ub also has a higher affinity. We here use di-Ub as we are specifically interested in the interactions with the allosteric site in the RING1-IBR helix and the effect of the HOIL-1 I358R mutant. Di-Ub provides the best balance between a quantifiable affinity and specificity around interactions with I358.

As discussed in response to point #2 above, we also observe a relatively large apparent discrepancy between the K_D determined by ITC and the EC_{50} determined in our biochemical E2-Ub discharge assays. In the latter, the EC_{50} for M1-linked di-Ub is 8 μ M (Fig. 1c) compared to the K_D of 50 μ M determined by ITC. For our system, the EC_{50} measured in activity assays seems the more relevant value.

7. Page 9, line 18: The authors speculate that D384 and D385 of HOIL-1 may contribute to the binding to the allosteric ubiquitin. It should be possible to generate a triple mutant - E383A/D384A/D385A - to verify such a statement.

RESPONSE: We now generated the E383A/D384A/D385A mutant to test our previous hypothesis as suggested by the reviewer. Different to our hypothesis in the initial manuscript, we did not observe a difference in the allosteric activation between HOIL-1 E383A and this triple mutant, suggesting interactions between the Ub C-terminal Arg residues and this negatively charged patch in HOIL-1 are not contributing to the allosteric activation. This is different to the importance of a similar interaction in HOIP, where mutation of the analogous Glu809 reduces allosteric activation (Lechtenberg *et al.*, Nature, 2016). We include this experiment in new Supplementary Fig. 5c and revised the main text on p. 9 to reflect these new results. Importantly, the effects of the I358R mutant in the biochemical and ITC assays support the Ub interaction observed in our crystal structure and the general effect of allosteric activation of HOIL-1.

8. Figure 2: The tabulated ITC result of HOIL-1/M1 di-Ub has two entries with different K_D values, 0.5 and 0.2 μ M. What is the second one that is not described in the manuscript? Additionally, several molar ratios (n values) clearly deviated from unity, 1.22, 0.71 and 0.78. Could this be inaccurate protein concentration estimation or significant proportions of inactive binding partners?

RESPONSE: We performed the ITC experiment of HOIL-1/M1 di-Ub twice under the same conditions, and we therefore report both experimental values in the table. Also, in response to Minor comment 1 below, we now report both values in the main text as a range, 0.2 – 0.5 μ M, rather than the average. All other titrations were performed as single titrations, partly due to the high protein consumption of ITC, especially of the relatively time-consuming to generate UbcH7(C86K)-Ub conjugate. However, throughout the development of the method and evolution of the study, we have performed multiple titrations under slightly different conditions with comparable results, giving us confidence in the reliability of the data.

Based on similar titrations with other E3 ligases in the lab (unpublished) with which we have determined a molar ratio of 1, we believe that the deviation from $n=1$ molar ratio is due to inaccuracies in the “active” (i.e., able to bind UbcH7-Ub) concentration of the E3 ligase. This may be due to inaccuracies in determining the overall protein concentration. We measured absorbance at 280 nm using a Nanodrop Microvolume Spectrophotometer (Thermo Fisher) and calculated protein concentrations using extinction coefficients determined from the amino acid sequence by the ExPASy ProtParam tool (<https://web.expasy.org/protparam/>). We also cannot exclude a small fraction of “inactive” protein due to effects such as protein aggregation or precipitation. We aim to minimise these effects by snap freezing proteins in liquid nitrogen in small aliquots (< 100 μ l), storing thawed proteins on ice, using proteins within short time frames after thawing (ideally immediately, but always within the same day), and avoiding freeze-thaw cycles as much as possible.

It is important to note, that despite these differences in the molar ratio in the two titrations with HOIL-1 ($n = 1.22$ and $n = 0.99$, respectively), we only observe small changes in the binding parameter ΔH , $-\Delta S$ and K_d that do not affect our conclusions. We therefore are confident that the deviations in the other titrations do not affect our overall results and conclusions.

9. The SDS-PAGE of Figure 1d exhibits significant amount of high-molecular-weight material in HOIL-1 helix-RBR that resembles NZF-RBR. In the reconstitute biochemical assays, the individual protein components are expected to be highly purified. What can be the high-molecular-weight material in this SDS-PAGE?

RESPONSE: The band appearing just under the 37 kDa marker bands in Fig. 1d is present in all E2-Ub discharge assays with HOIL-1 and RNF216 and is a Ubch7 dimer. This Ubch7 dimer is most likely a disulfide linked dimer as it is present in gels run under non-reducing conditions (most gels in our manuscript, in particular all E2-Ub thioester discharge assays) but disappears under reducing conditions (see e.g., Fig. 7e). While it is unclear if this disulfide linked Ubch7 species is present during our experiments or only forms under the high protein concentrations during SDS-PAGE, in any case it is a minor species of the total Ubch7 present in our system, and we do not think it affects interpretation of our results. The band intensity increases over the time-course of the experiment as the ubiquitin-charged Ubch7 dimer (visible e.g., in Supplementary Fig. 5f) is discharged over time.

We now collated all uncropped gels in the Source Data File, where the E2 and E2-Ub homodimers as well as the E2/E2-Ub heterodimer are more evident. Figure 1 for reviewers below is a fully annotated uncropped gel of Fig. 1d, better visualising these bands.

Figure 1 for reviewers: Uncropped gel of the HOIL-1 Ubch7-Ub discharge assay presented in Fig. 1d of the manuscript. The additional Ubch7-Ub (E2-Ub) dimer bands are labelled.

10. Page 6, last paragraph and Supplementary Fig. 3e: What could be the contributing factors of spontaneous degradation of M1 di-Ub into mono-ubiquitin in the crystals of HOIL-1/Ubch7~Ub/M1 di-Ub?

RESPONSE: We think that the di-Ub degradation is due to a very minor contamination with a protease of unknown source. We observe that after 4 days at room temperature, some M1 di-Ub is

still present (Supplementary Fig. 3e), suggesting that this is a minor contamination that does not affect our biochemical or biophysical experiments.

Minor comments:

1. Page 4, line 32: An error estimate of the dissociation constant for HOIL-1 binding to Ubch7(C86K)-Ub, stimulated by M1 di-Ub, was provided, $K_D = 0.35 \pm 0.15 \mu\text{M}$, but $n=2$. If the error estimate is RMSD, the minimum sample size should be 3.

RESPONSE: The error given is the standard error of the mean (SEM). We revised this sentence and now provide the K_D of the two experiments as the range, 0.2 – 0.5 μM , which provides a more accurate description of the experiments. We also added a description to figure legend 2c to explain that the titration of Ubch7-Ub into HOIL-1/M1 di-Ub was performed in duplicate.

2. Page 4, line 29: A mutant of Ubch7(C86K) is used in the binding assay instead of wild-type Ubch7, but the corresponding labels in Fig. 2 do not specify the C86K mutation (but the corresponding caption also uses the same description, Ubch7(C86K)). The nomenclature should be used consistently or defined in the manuscript when abbreviated/simplified forms are used.

RESPONSE: The details of the mutant are specified in the figure legend, but we acknowledge that it is best practice to include as much information as possible in the actual figure. We have now revised the labels and text throughout the manuscript to better indicate which mutants are used in places where this is critical while maintaining readability.

3. di-Ub is labeled as Di-Ub in Figure 1, Supplementary Figure 2c (Table) and other places.

RESPONSE: We revised Supplementary Figure 2 and one occurrence of a mid-sentence “Di-Ub” in the Methods section. Otherwise, labels in our figures and Supplementary Tables follow the *Nature Communications* style guide that specifies: “Lettering in figures should be in lower-case type, with only the first letter of each label capitalized.”

4. Page 2, line 32: NEDD should be capitalized.

RESPONSE: Thanks, fixed.

Reviewer #3:

Remarks to the authors:

Wang et al investigate the catalytic mechanism of the RING-between-RING (RBR) E3 ligases HOIL-1 and RNF216. These enzymes transfer ubiquitin from the loaded E2 conjugating enzyme to substrate via a covalently linked intermediate. A common feature of these enzymes is that they generate ubiquitin chains linked in a defined fashion and these diubiquitin (diUb) linked products tend to stimulate ubiquitination in a feed-forward reaction. At the outset the authors state that the precise mechanism by which ubiquitin binding at an allosteric site enhances catalysis has not been explored in detail. The authors attempt to answer this question using HOIL-1 and RNF216 and demonstrate that M1 linked diUb and K63 linked diUb respectively act as activators of the first step in the enzymatic cascade, stimulating binding of the ubiquitin loaded E2 (Ubch7~Ub) to a fragment of the RBR and accelerating transfer of the Ub to a cysteine residue in the E3 (transthiolation). Unlike the classical RING transfer mechanism the Ubch7~Ub is stretched out in an “open” conformation rather than in a folded back or “closed” conformation.

While the structural and biochemical experiments are well done and in general support any conclusions drawn, I feel that the authors have not really answered the question posed at the start

of the paper namely “ how does allosteric diUb activate the transthiolation reaction”. In both HOIL-1 and RNF216 only a single Ub is seen bound to the RBR E3, even although in the case of HOIL-1 M1 linked diUb was used for structural determination (although there was some breakdown to monoUb during crystallography). The difficulty here is that the biochemical reactions are activated by specific diUb linkages (and not monoUb) and if we can't see how the diUb species are bound we can't know how they activate the reaction. Binding of the monoUb is to the backside of the RBR and while this appears to straighten out a helix N-terminal to RING1, it is not clear how this contributes to binding of the Ub_{CH7}~Ub in a conformation that allows the active site cysteine of the RBR to attack the thioester of the Ub_{CH7}~Ub. This is really the key question and it has not been answered.

RESPONSE: We thank the reviewer for their positive comments on the quality and rigor of our experiments. The main rationale for our study is to identify the common features of the RBR mechanism and deviations thereof, which may point to unique functions of specific RBR family members. We addressed this rationale by solving the structures of the transthiolation complexes of the previously less-well studied RBRs HOIL-1 and RNF216. In our view, and as outlined throughout our manuscript, our structures, together with complementary biochemical and biophysical work, and in the context of previous studies on the better understood RBRs HOIP, Parkin and HHARI, clearly point to the defining features of the RBR mechanism. Our work further identifies unique adaptations, most prominently the Zn₂/Cys₆ bivalent zinc cluster replacing part of HOIL-1's RING2 domain. While we were unable to fully define the function of this zinc cluster in our current study, it provides the foundation for future more targeted studies by us and others in the field.

The reviewer is correct that allosteric activation by (di-)Ub is one question we aim to address in our work. In response to this reviewer and comments by the other reviewers, we now include new data in Supplementary Figure 5d that show that HOIL-1 is weakly activated by mono-Ub at a very high concentration. The activity of HOIL-1 in presence of 200 μM mono-Ub is only 20% of that of HOIL-1 in the presence of 40-fold less (5 μM) M1-linked di-Ub, but it is 2-times more active than HOIL-1 in the absence of an activator. These new data together with our crystal structures and our mutation experiments in which we mutate the Ile44 patches of the proximal and/or distal Ub of M1-linked di-Ub (now moved to Supplementary Fig. 5e) in our view support a model in which the Ub protomer binding to the allosteric site in the RING1-IBR helix as observed in our crystal structure induces the conformational changes in the RBR that lead to enhanced catalytic activity. The 2nd (and potential additional) Ub molecule is important to enhance binding avidity and at the same time introduces chain-type specificity of the activation.

Regarding the mechanism, our current working model is that straightening of the RING1-IBR helix re-arranges or stabilises the RBR in a conformation that enhances binding to E2-Ub conjugate (supported by our ITC experiments, Fig 2) by forming a binding site for the donor Ub on the other side of the helix and supporting alignment of the RING1 and RING2 domains to bind the E2-Ub conjugate. RBR ligases are highly dynamic proteins that undergo vast conformational changes through their activation process and catalytic cycle as shown by multiple studies using HDX-MS (Gladkova *et al.*, Nature 2018; Sauve *et al.*, NSMB 2018) and X-ray/cryo-EM structures in different activation states (see e.g., our Supplementary Fig. 8).

One of the novel features of this paper is the observation that the HOIL-1 RING2 domain comprises a Zn₂/Cys₆ binuclear cluster that replaces the C-terminal half of RING2. A recently discovered feature of HOIL-1 is that it has the ability to ubiquitinate polysaccharides and the authors suggest that the Zn₂/Cys₆ cluster is involved in this reaction. However this argument is somewhat undermined by the observation that RNF216 does not have the Zn₂/Cys₆ cluster, but is still capable of ubiquitinating maltose.

RESPONSE: We fully agree with the reviewer that the identification of the Zn²⁺/Cys6 binuclear cluster in the HOIL-1 RING2 domain is an exciting finding. It is also correct that given that this cluster is not found in any other (RBR) E3 ligase, and given the recently reported atypical HOIL-1 substrate specificity, we hypothesized that the Zn²⁺/Cys6 binuclear cluster may be critical for HOIL-1 polysaccharide specificity. We tested this hypothesis in our biochemical experiments, and while we confirmed that HOIL-1 can ubiquitinate the polysaccharide maltose (as reported previously), our novel control experiments using other RBRs (RNF216 and HOIP) that do not contain the Zn²⁺/Cys6 binuclear cluster also showed activity towards maltose, therefore not supporting our initial hypothesis. As outlined in the previous response, our manuscript mainly focuses on the general RBR mechanism, and while we believe that more detailed studies on the function of the HOIL-1 Zn²⁺/Cys6 binuclear cluster are required (and are in fact ongoing in our laboratory), these are outside the scope of the current manuscript. Nevertheless, we believe these results are important for the field as they put the recent finding by Kellsall *et al.* (*EMBO J* 2022, ref. 24) into perspective and provide the structural basis for subsequent work in this field.

In general the paper is well done and there are novel aspects to it, although I feel that it needs to go a bit further in answering its central question.

RESPONSE: We again thank the reviewer for their positive comments on the quality and novelty of our work and hope that our responses to all reviewers throughout this document and additional data have alleviated the reviewer's remaining concerns.

Reviewer #4 (Remarks to the Author):

The manuscript by Wang *et al.* describes a structural and biochemical workup of two Ring-between-Ring (RBR) ubiquitin ligases, HOIL-1 and RNF216. RBRs are unique ubiquitin ligases in that they catalyze ubiquitylation by first binding to an E2~ubiquitin molecule and promoting the transfer of ubiquitin from the ubiquitin-conjugating enzyme E2 to the RING 2 domain of the ligase, and then ultimately to a protein substrate bound to the ligase. Several RBRs including PARKIN, ARIH1(HHARI) and HOIP have had their molecular mechanisms described in detail, whereas those of HOIL-1 and RNF216 still awaited discovery and was the main thrust of this paper. X-ray crystal structures of fragments for both ligases are presented and buttressed by a suite of biochemical experiments ranging from isothermal calorimetry, size exclusion chromatography coupled with multi-angle light scattering, and numerous kinetic ubiquitylation assays. The results show that important aspects of HOIL-1 and RNF216 function, including E2~ubiquitin binding to the ligase, transfer of ubiquitin from E2 to the RING2 domain, and the binding of an allosteric ubiquitin or ubiquitin-like activator, are similar as existing models for RBRs in the literature. Specifically, the addition of free di-ubiquitin results in the stimulation of E2~ubiquitin binding to both ligases and in a linkage-specific manner. Mutants of both structures were biochemically characterized to confirm various aspects of the model. While HOIL-1 and RNF216 appear to function similarly as the other RBRs mentioned here, differences were also observed in the details and are described in the manuscript. A model is presented where protein substrate bound to RBR becomes modified with a poly-ubiquitin chain which then allosterically enhances ligase activity through a feed-forward mechanism.

This well-written paper was a pleasure to read and is packed with useful information regarding RBR ubiquitin ligase function. While the results lead to models that are clearly similar to those in the literature for better characterized E3s, this should not take away from the impact of the work, as our understanding of RBRs in general is still lacking. As such, this study is a major advance and should be published in *Nature Communications*, though we suggest the following changes prior to that.

Major points:

While the feed forward mechanism is interesting, there are a couple of aspects that make the model challenging to accept, at least for now. First, the protein substrate must receive at minimum 2 ubiquitins to generate the di-ubiquitin chain, although since the substrate binding site is not mentioned or shown, it very well may be the case that far more ubiquitins are necessary for the chain to reach the allosteric site. As such, it is unclear what the functional advantage would be in a case where several ubiquitins are first transferred to protein substrate prior to engagement of the allosteric mechanism for enhanced activity. Another issue with the model is that the authors show here that E2~ub binding is weak in the absence of the allosteric activator, perhaps even too weak for E2~ub to associate with the RBR-substrate complex in cells where UBCH7 levels are relatively low (indeed the authors were not even able to measure an equilibrium dissociation constant for E2 and HOIL-1 in the absence of di-ubiquitin). As such, perhaps this detailed model should be removed from the manuscript unless far more substantial evidence can be produced.

RESPONSE: We thank the reviewer for raising these important points and allowing us to clarify some aspects of our model that we did not describe clearly enough. First, the allosteric ubiquitin may not be part of the same ubiquitin chain as the acceptor ubiquitin, i.e., allosteric di-Ub and acceptor Ub in the active site can come from different chains. This is well appreciated for the RBR Parkin and its activator pUb. Research has shown that pUb mostly caps short Ub chains (Swatek et al., Nature 2019) and is a poor substrate of Parkin (Wauer et al., EMBO J, 2015; Ordureau et al., Mol Cell, 2014). In the case of HOIL-1, HOIL-1 is activated by M1- or K63-linked chains, which are not chain types formed by HOIL-1, but instead likely are formed by HOIL-1's interaction partner HOIP (in the case of M1 chains) or other E3 ligases associated with the LUBAC and its signalling pathways.

We strongly believe that our model adds value to the manuscript and prefer to keep it. However, we acknowledge that parts of the model are not (yet) well supported by experimental evidence. We acknowledge this by using '?' in the model and now change the title of Figure 8 from "General model of the RBR mechanism" to "Working model of the RBR mechanism". We now also specify that the model is based on our current work as well as previously published work by us and others in the figure legend. We hope that these clarifications and changes now make the model more acceptable to the reviewer.

Minor points:

Supplementary Figure 1b: the experiment is designed to follow the effect of the presence of di-ubiquitin on the stimulation of HOIL-1 thioester bond formation with ubiquitin. However, we wonder how the authors distinguish this from isopeptide bond formation between HOIL-1 and ubiquitin. Comparison of reducing and non-reducing gels would be helpful here.

RESPONSE: The experiments in Supplementary Figure 1 were performed with HOIP rather than HOIL-1 for the reasons described in our manuscript (p.5, l. 14). The HOIP-Ub thioester formation assay in Supplementary Figure 1b is based on our previous work (Lechtenberg *et al.*, Nature, 2016). In this previous work, in Supplementary Figure 1 – Western blot for Figure 3d (reproduced below as Figure 2 for reviewers), we show that the HOIP-Ub band fully disappears when the SDS-PAGE is run under reducing conditions (with DTT added to the sample). We are therefore confident that the HOIP-Ub represents the HOIP-Ub thioester intermediate rather than a HOIP-Ub isopeptide product.

Figure 3d

Figure 2 for reviewers (reproduced from Lechtenberg et al, 2016, Nature; Supplementary Figure 1). HOIP-Ub thioester formation assay run under non-reducing (- DTT) and reducing (+ DTT) conditions, showing that the HOIP-Ub band represents the HOIP-Ub thioester and not isopeptide linked ubiquitinated HOIP. Note: The readout used in this figure is an anti-Ub (P4D1) Western blot rather than Coomassie staining used in the current manuscript.

Figure 7e: it is speculated that the mutation of His 510 to Ala in HOIL-1 stabilizes the thioester bond between the mutant ligase and ubiquitin as evidenced by the appearance of a slowly migrated band in the gel. While this is compelling, the presence of doublet bands makes interpretation of the result more challenging (in other words, if this really is HOIL-1~cys-ubiquitin, why are there two bands that disappear upon treatment with reducing agent). Also, it would be helpful to have both a non-reducing and reducing gel for the WT ligase for the sake of comparison.

RESPONSE: The 2nd band that disappears upon treatment with reducing agent likely represents a Ubch7 dimer that is present on all our gels run under non-reducing conditions. We discussed this in more detail in response to reviewer 2, major point 9, above. We now include gels run under non-reducing and reducing conditions for both WT and the H510A mutant in Fig 7e. We also include a supplement with uncropped gels that more clearly show these Ubch7 dimers.

A defect is reported for H510A HOIL-1 in maltose ubiquitylation. However, this is potentially a non-physiological substrate and as such, its relevance to HOIL-1 function is still poorly understood. We suggest addressing this by adding an experiment comparing the activity of H510A to wild-type in the aminolysis assay with TAMRA labeled ubiquitin.

RESPONSE: HOIL-1 activities involving different substrates have been proposed in the literature, including Lys-linked autoubiquitination, Ser/Thr-linked autoubiquitination, Ser/Thr-linked di-Ub formation and, most recently, polysaccharides including maltose. At this stage it is an open question which of these are truly physiological substrates, and more detailed biological studies are required to comprehensively answer this. This is ongoing work in our lab, but outside the scope of this study.

In our initial work, we focused on maltose, as we hypothesised that the HOIL-1 Zn²/Cys6 binuclear cluster may explain HOIL-1's potential unique activity.

Nevertheless, we attempted the experiment proposed by the reviewer, using TAMRA-Ub as an acceptor (see Figure 3 for reviewers below). While we see robust ubiquitination of maltose with WT HOIL-1 but not the H510A mutant, we do not observe formation of di-Ub with either HOIL-1 species. We do observe HOIL-1 autoubiquitination with both WT and H510A mutant, likely because of the non-specific reactivity of the charged HOIL-1-Ub thioester with Lys/Ser/Thr residues in the vicinity of the HOIL-1 active site. As we observed previously in Fig. 7e,f, the autoubiquitination is stronger with the H510A mutant. Given these results, we think that the maltose ubiquitination assay shown in our manuscript represents the most relevant substrate assay.

During the revision of our manuscript, the Stieglitz/Rittinger labs posted a manuscript on the BioRxiv preprint server (<https://doi.org/10.1101/2022.11.13.516300>). In this preprint, they also identify HOIL-1 H510 as a catalytic residue. While they use a less direct activity assay (see Fig. 1 E–H of that preprint) that looks at how HOIL-1 activity affects activity of the E3 ligase HOIP, they conclude that H510 is a catalytic residue, in line with our work.

Figure 3 for reviewers. HOIL-1 activity assay using TAMRA-Ub. a HOIL-1 substrate ubiquitination assay with WT and H510A HOIL-1. The left panel shows a Coomassie-stained gel, the right panel the fluorescence signal of the same gel. We observe HOIL-1 autoubiquitination, but no di-Ub formation. **b** Maltose ubiquitination assay using TAMRA-Ub, showing that TAMRA-Ub is active with HOIL-1 and can still be conjugated to maltose.

Reviewer #1 (Remarks to the Author):

The authors have provided new experiments that addressed my concerns. I support the publication of the manuscript.

Reviewer #2 (Remarks to the Author):

The authors have sufficiently address my comments to the original manuscript with additional experiments that actually lead to different interpretations of the original hypothesis, which the authors also clarified in the manuscript. I do not have further comment to make, and I recommend the publication of the revised manuscript in Nature Communications.

Reviewer #3 (Remarks to the Author):

I think ther authors have made a pretty good attempt to address the points raised in my review. The new manuscript adds to our understanding of the mechanism of RING between RING ubiquitin E3 ligases and is now suitable for publication.

Reviewer #4 (Remarks to the Author):

No comments